# Efficient Mirror Descent Ascent Methods for Nonsmooth Minimax Problems

**Feihu Huang, Xidong Wu, Heng Huang**
Department of Electrical and Computer Engineering, University of Pittsburgh, Pittsburgh, USA
`huangfeihu2018@gmail.com, xidong-wu@pitt.edu, heng.huang@pitt.edu`

## Abstract

In the paper, we propose a class of efficient mirror descent ascent methods to solve the nonsmooth nonconvex-strongly-concave minimax problems by using dynamic mirror functions, and introduce a convergence analysis framework to conduct rigorous theoretical analysis for our mirror descent ascent methods. For our stochastic algorithms, we first prove that the mini-batch stochastic mirror descent ascent (SMDA) method obtains a gradient complexity of $O(\kappa^3\epsilon^{-4})$ for finding an $\epsilon$-stationary point, where $\kappa$ denotes the condition number. Further, we propose an accelerated stochastic mirror descent ascent (VR-SMDA) method based on the variance reduced technique. We prove that our VR-SMDA method achieves a lower gradient complexity of $O(\kappa^3\epsilon^{-3})$. For our deterministic algorithm, we prove that our deterministic mirror descent ascent (MDA) achieves a lower gradient complexity of $O(\sqrt{\kappa}\epsilon^{-2})$ under mild conditions, which matches the best known complexity in solving smooth nonconvex-strongly-concave minimax optimization. We conduct the experiments on fair classifier and robust neural network training tasks to demonstrate the efficiency of our new algorithms.

## 1 Introduction

Minimax optimization recently has attracted increased interest largely due to advance in many machine learning applications such as generative adversarial networks (GANs) [14, 41], robust neural networks training [32], fair learning [31], federated learning [10], and policy evaluation [43]. In the paper, we study the following nonsmooth nonconvex-strongly-concave minimax problem:

$$\min_{x\in\mathcal{X}}\max_{y\in\mathcal{Y}} F(x,y) = \big\{f(x,y) + g(x) - h(y)\big\}, \tag{1}$$

where the function $f(x,y) : \mathcal{X}\times\mathcal{Y} \to \mathbb{R}$ is smooth and possibly nonconvex in $x \in \mathcal{X}$ and $\mu$-strongly concave in $y \in \mathcal{Y}$, and the functions $g(x)$ and $h(y)$ are convex and possibly nonsmooth. Here $\mathcal{X} \subseteq \mathbb{R}^d$ and $\mathcal{Y} \subseteq \mathbb{R}^p$ are compact and convex constraint sets, or $\mathcal{X} = \mathbb{R}^d$ and $\mathcal{Y} = \mathbb{R}^p$. In many machine learning problems, $f(x,y)$ generally represents loss function and is a stochastic form, *i.e.*, $f(x,y) = \mathbb{E}_\xi[f(x,y;\xi)]$, where the random variable $\xi$ follows an unknown data distribution. Here both $g(x)$ and $h(y)$ frequently denote the nonsmooth regularization terms such as $g(x) = \nu_1\|x\|_1$ and $h(y) = \nu_2\|y\|_1$ with $\nu_1, \nu_2 > 0$. In fact, the above problem (1) comes from many machine learning problems, such as fair classifier, robust training, nonlinear temporal-difference learning in reinforcement learning [43] and robust federated learning [10].

When $g(x) = 0$ and $h(y) = 0$ in the problem (1), the classic gradient descent ascent (GDA) methods [40, 27] can effectively solve this problem, which alternatively conducts a gradient descent update on the variable $x$ and a gradient ascent update on the variable $y$ at each iteration. At the same time, some stochastic GDA methods [40, 27, 30, 18, 46, 17] have been proposed to solve the stochastic minimax problem (1), where $f(x,y) = \mathbb{E}_\xi[f(x,y;\xi)]$. More recently, some works [2, 4, 8] focus on more general minimax problem (1), where both $g(x)$ and $h(y)$ are possibly nonsmooth. Meanwhile,

35th Conference on Neural Information Processing Systems (NeurIPS 2021).

Table 1: Gradient complexity of the representative first-order methods for obtaining an $\epsilon$-stationary point of the **nonsmooth nonconvex** minimax problem (1). Note that these comparison methods don't rely on some specific strong assumptions on the problem (1). Since the convergence properties of the deterministic proximal gradient descent ascent (PGDA) [8] build on the Kurdyka-Lojasiewicz (KL) geometry assumption, it be excluded. Here $\kappa$ denotes condition number of objective function $f(x, y)$ in variable $y$. Since HiBSA algorithm [29] does not provide explicit dependence on $\kappa$, we use $p(\kappa)$.

| Type | Algorithm | Reference | Loop(s) | Gradient Complexity |
|---|---|---|---|---|
| Deterministic | HiBSA | [29] | Single | $O(p(\kappa)\epsilon^{-2})$ |
| | MAPGDA | [2] | Double | $O(\kappa^{3/2}\epsilon^{-2})$ |
| | PAGDA | [4] | Single | $O(\kappa^2\epsilon^{-2})$ |
| | MDA | Ours | Single | $O(\sqrt{\kappa}\epsilon^{-2})$ |
| Stochastic | PASGDA | [4] | Single | $O(\kappa^3\epsilon^{-4})$ |
| | SMDA | Ours | Single | $O(\kappa^3\epsilon^{-4})$ |
| | VR-SMDA | Ours | Double | $O(\kappa^3\epsilon^{-3})$ |

some (stochastic) proximal gradient descent ascent (PGDA) methods [2, 4, 8] have been presented to solve the problem (1). However, they still suffer from the large sample complexities for finding an stationary point of the minimax problem (1) without some specific strong assumptions such as KL geometry (Please see Table 1).

In this paper, thus, we propose a class of efficient mirror descent ascent methods by using dynamic mirror function (i.e., Bregman function). Specifically, our methods perform an adaptive mirror descent update to variable $x$ and an adaptive mirror ascent update to variable $y$ alternatively at each iteration. Our new algorithmic framework can generate many popular methods and their variants by adopting different mirror functions. For example, by adopting the mirror functions $\psi(x) = \frac{1}{2}\|x\|^2$ and $\phi(y) = \frac{1}{2}\|y\|^2$, our methods will include the classic (proximal) gradient descent ascent algorithms. Our main contributions are summarized as follows:

1) We propose a class of novel mirror descent ascent methods to solve the minimax problem (1) by using dynamic mirror functions. Moreover, we provide a convergence analysis framework for our mirror descent ascent methods.

2) We present a faster **deterministic** adaptive mirror descent ascent (MDA) method, which reaches a lower gradient complexity of $O(\sqrt{\kappa}\epsilon^{-2})$ than the existing nonsmooth nonconvex minimax methods. Meanwhile, we propose a fast **stochastic** mirror descent ascent (SMDA) method, which requires $O(\kappa^3\epsilon^{-4})$ stochastic gradient evaluations to obtain an $\epsilon$-stationary point of the problem (1).

3) We further propose an accelerated **stochastic** mirror descent ascent (VR-SMDA) method by using the variance reduced technique of SARAH/SNVRG/SPIDER [37, 51, 12, 44]. Moreover, we prove that our VR-SMDA reaches a lower gradient complexity of $O(\kappa^3\epsilon^{-3})$.

In fact, when our methods solve the minimax problem (1) without nonsmooth regularization terms, i.e., $g(x) = 0$ and $h(y) = 0$, our theoretical results also can apply these minimax problems without nonsmooth regularization terms studied in [27, 28].

## 2 Related Works

In this section, we review some existing typical minimax optimization methods and stochastic mirror descent methods, respectively.

### 2.1 Minimax Optimization Methods

Minimax optimization recently has been widely studied in machine learning community. The convergence properties of (strongly) convex-(strongly) concave minimax optimization have been studied in [42, 26, 35, 49]. Due to the popularity of nonconvex models in machine learning, many recent studies focused on the nonconvex minimax problems such as robustly deep neural networks training and GANs. For example, some effective gradient descent ascent (GDA) methods [40, 38, 27, 28, 46, 47, 17, 18, 19, 16] have been proposed for the nonconvex-(strongly) concave minimax optimization. Specifically, Lin et al. [27] studied the convergence properties of both deterministic and stochastic GDA methods. Subsequently, Luo et al. [30] proposed a class of faster stochastic GDA methods based on variance reduced technique of SPIDER [12]. Huang et al. [18] proposed

an accelerated single-loop stochastic GDA method based on momentum-based variance reduced technique of STORM [9]. Recently, the nonconvex-nonconcave minimax optimization problems with some specific structures also have been studied in [34, 48, 11, 33]. Specifically, Yang et al. [48] and Diakonikolas et al. [11] have studied a class of specific nonconvex-nonconcave minimax problems satisfying a so-called two-sided Polyak-Lojasiewicz inequality and stampacchia variational inequality, respectively. More recently, Zhang et al. [50] and Li et al. [24] studied the lower bound of sample complexity for nonconvex-strongly-concave minimax optimization. Xian et al. [45] studied the decentralized nonconvex-strongly-concave minimax optimization. Meanwhile, some research works [29, 2, 4, 8] began to study the nonsmooth nonconvex minimax problem (1).

## 2.2 Mirror Descent Methods

Mirror descent (a.k.a., Bregman gradient) method [7, 3] is a powerful optimization tool in machine learning, since it can fit the geometry of optimization problems by choosing proper Bregman functions [5, 6]. The mirror descent methods for convex optimization have been studied in [7, 3]. Subsequently, Lei et al. [23] integrated the variance reduced technique to the mirror descent algorithm for stochastic convex optimization. More recently, a variance-reduced adaptive stochastic mirror descent algorithm [25] has been proposed to solve the nonsmooth nonconvex finite-sum mini optimization. Recently, the mirror descent method also has been used to solve minimax optimization problems. For example, Babanezhad et al. [1] presented a mirror-type algorithm for convex minimax optimization. Rafique et al. [40] proposed a class of mirror descent methods for weakly convex minimax optimization. Meanwhile, a new mirror descent-type method [34] has been proposed to solve a class of nonconvex-nonconcave minimax problems with a non-monotone variational inequality structure. To the best of our knowledge, recently few work focuses on explicitly using the mirror-decent-type method to solve the nonsmooth nonconvex minimax problems.

# 3 Preliminaries

## 3.1 Notations

For two vectors $x$ and $y$, $\langle x, y \rangle$ denotes their inner product. $\| \cdot \|$ denotes the $\ell_2$ norm for vectors and spectral norm for matrices, respectively. $\nabla_x f(x, y)$ and $\nabla_y f(x, y)$ denote the partial derivatives *w.r.t.* variables $x$ and $y$ respectively, and let $\nabla f(x, y) = \big( \nabla_x f(x, y), \nabla_y f(x, y) \big)$. $\partial g(x)$ denotes the subgradient set of function $g(x)$. Given a convex closed set $\mathcal{X}$, we define a projection operation $\mathrm{Proj}_{\mathcal{X}}(\alpha) = \arg\min_{x \in \mathcal{X}} \|x - \alpha\|^2$. Given the mini-batch samples $\mathcal{B} = \{\xi^i\}_{i=1}^b$, we let $\nabla f_{\mathcal{B}}(x) = \frac{1}{b} \sum_{i=1}^b \nabla f(x; \xi^i)$. Define an increasing $\sigma$-algebras $\mathcal{F}_t := \{\mathcal{B}_1, \mathcal{B}_2, \cdots, \mathcal{B}_{t-1}\}$ for all $t \geq 2$, then let $\mathbb{E}[\cdot] = \mathbb{E}[\cdot | \mathcal{F}_t]$.

## 3.2 Standard Mirror Descent Method

Given a $\rho$-strongly convex and continuously-differentiable function $\psi(x)$, *i.e.*, $\langle x_1 - x_2, \nabla\psi(x_1) - \nabla\psi(x_2) \rangle \geq \rho \|x_1 - x_2\|^2$, we define a Bregman divergence (i.e., Bregman distance) for any $x, z \in \mathcal{X}$:

$$D_\psi(z, x) = \psi(z) - \psi(x) - \langle \nabla\psi(x), z - x \rangle. \tag{2}$$

To solve the problem $\min_{x \in \mathcal{X}} f(x)$, the mirror descent method [7, 3] uses the following form to update the variable $x$, for all $t \geq 1$

$$x_{t+1} = \arg\min_{x \in \mathcal{X}} \left\{ f(x_t) + \langle \nabla f(x_t), x - x_t \rangle + \frac{1}{\alpha} D_\psi(x, x_t) \right\}, \tag{3}$$

where $\alpha > 0$ is stepsize. In the above subproblem (3), the first two terms of its objective function is a linear function approximated the function $f(x)$, and the last term is a Bregman distance between $x$ and $x_t$. Note that the constant terms $f(x_t)$ and $\langle \nabla f(x_t), x_t \rangle$ can be omitted in the above subproblem (3). When choosing $\psi(x) = \frac{1}{2}\|x\|^2$, we have $D_\psi(x, x_t) = \frac{1}{2}\|x - x_t\|^2$. Then the mirror descent method will reduce to the standard projected gradient descent method.

## 3.3 Some Mild Assumptions

In the subsection, we introduce some mild assumptions for the problem (1).

**Assumption 1.** *(Smoothness) For the deterministic and mini-batch stochastic algorithms (MDA and SMDA), we assume that the function $f(x, y)$ has an $L_f$-Lipschitz gradient, i.e., for all $x_1, x_2 \in \mathcal{X}$ and $y_1, y_2 \in \mathcal{Y}$, we have*

$$\|\nabla f(x_1, y_1) - \nabla f(x_2, y_2)\| \leq L_f \|(x_1, y_1) - (x_2, y_2)\|. \tag{4}$$

*For our variance-reduced stochastic algorithm (VR-SMDA), we assume that each component function $f(x, y; \xi)$ has an $L_f$-Lipschitz gradient, i.e., for all $x_1, x_2 \in \mathcal{X}$ and $y_1, y_2 \in \mathcal{Y}$, we have*

$$\|\nabla f(x_1, y_1; \xi) - \nabla f(x_2, y_2; \xi)\| \leq L_f \|(x_1, y_1) - (x_2, y_2)\|, \ \forall \xi. \tag{5}$$

In Assumption 1, the inequality (4) is commonly used in the minimax optimization [27, 2, 4, 8]. While the inequality (5) is frequently used in the variance-reduced stochastic optimization [30, 18].

**Assumption 2.** *Each component function $f(x, y; \xi)$ has an unbiased stochastic gradient with bounded variance $\sigma^2$, i.e.,*

$$\mathbb{E}[\nabla f(x, y; \xi)] = \nabla f(x, y), \quad \mathbb{E}\|\nabla f(x, y; \xi) - \nabla f(x, y)\|^2 \leq \sigma^2. \tag{6}$$

**Assumption 3.** *The function $f(x, y)$ is $\mu$-strongly concave w.r.t $y$, i.e., for all $x \in \mathcal{X}$ and $y_1, y_2 \in \mathcal{Y}$, we have $\|\nabla_y f(x, y_1) - \nabla_y f(x, y_2)\| \geq \mu \|y_1 - y_2\|$. Then the following inequality holds*

$$f(x, y_1) \leq f(x, y_2) + \langle \nabla_y f(x, y_2), y_1 - y_2 \rangle - \frac{\mu}{2} \|y_1 - y_2\|^2. \tag{7}$$

**Assumption 4.** *The functions $g(x)$ and $h(y)$ are convex but possibly nonsmooth.*

Assumption 3 shows that the function $f(x, y)$ is $\mu$-strongly concave w.r.t $y$. Assumption 4 shows that the function $h(y)$ is convex. Thus, the function $\{f(x, y) - h(y)\}$ is strongly concave in $y \in \mathcal{Y}$, there exists a unique solution to the problem $\max_{y \in \mathcal{Y}} \{f(x, y) - h(y)\}$ for any $x$. Let $y^*(x) = \arg\max_{y \in \mathcal{Y}} \{f(x, y) - h(y)\}$, and $\Phi(x) = f(x, y^*(x)) - h(y^*(x)) = \max_{y \in \mathcal{Y}} \{f(x, y) - h(y)\}$.

**Assumption 5.** *For any $\alpha \in \mathbb{R}$, the sub-level set $\{x : \Phi(x) + g(x) \leq \alpha\}$ is compact. The function $\Phi(x) + g(x)$ is bounded below in $\mathcal{X}$, i.e., $F^* = \inf_{x \in \mathcal{X}} \{\Phi(x) + g(x)\} > -\infty$.*

Assumption 5 is frequently used in nonsmooth minimax optimization [8]. In fact, when $h(y) = c$ where $c$ is a constant, we can only assume the function $\Phi(x) + g(x)$ is bounded below in $\mathcal{X}$ instead of Assumption 5.

## 4 Mirror Descent Ascent Methods

In the section, we propose a class of novel mirror descent ascent methods to solve the problem (1). Specifically, we first propose a deterministic mirror descent ascent (MDA) method, and stochastic mirror descent ascent (SMDA) method. Then we further present an accelerated stochastic mirror descent ascent (VR-SMDA) using variance reduced technique of SPIDER [12, 44].

### 4.1 MDA and SMDA algorithms

When $f(x, y)$ is a deterministic function, we propose a deterministic mirror descent ascent (MDA) method to solve the deterministic problem (1). When $f(x, y) = \mathbb{E}_\xi[f(x, y; \xi)]$ is a stochastic function, we propose a stochastic mirror descent ascent (SMDA) to solve the stochastic problem (1). Specifically, Algorithm 1 shows the algorithmic framework of the MDA and SMDA algorithms.

In Algorithm 1, we use (stochastic) mirror decent to update variable $x$, and simultaneously use (stochastic) mirror ascent to update variable $y$. Specifically, at step 7 of Algorithm 1, we use the mirror descent to update $x$,

$$x_{t+1} = \arg\min_{x \in \mathcal{X}} \left\{ \langle v_t, x \rangle + \frac{1}{\gamma_t} D_{\psi_t}(x, x_t) + g(x) \right\} \tag{8}$$

$$= \arg\min_{x \in \mathcal{X}} \left\{ f(x_t, y_t) + \langle v_t, x - x_t \rangle + \frac{1}{\gamma_t} D_{\psi_t}(x, x_t) + g(x) \right\}. \tag{9}$$

In fact, we omit the constant terms $f(x_t, y_t)$ and $\langle v_t, x_t \rangle$ in the above subproblem (8). In the above subproblem (9), the first two terms of its objective function is a linear function approximated the function $f(x, y)$ based on (stochastic) derivative estimator $v_t$, and the third term is a Bregman distance between $x$ and $x_t$ based on Bregman function $\psi_t$. Since the function $g(x)$ is possibly nonsmooth, we

---
**Algorithm 1** (Stochastic) Mirror Descent Ascent Algorithm
---
1: **Input:** $T$, stepsizes $\{\gamma_t > 0, \lambda_t > 0, \eta_t \in (0,1]\}_{t=1}^T$, mini-batch size $b$ ;
2: **initialize:** $x_1 \in \mathcal{X}$ and $y_1 \in \mathcal{Y}$;
3: **for** $t = 1, 2, \ldots, T$ **do**
4:     **MDA:** Compute partial derivatives $v_t = \nabla_x f(x_t, y_t)$ and $w_t = \nabla_y f(x_t, y_t)$;
5:     **SMDA:** Generate randomly mini-batch samples $\mathcal{B}_t = \{\xi_t^i\}_{i=1}^b$ with $|\mathcal{B}_t| = b$, and compute stochastic partial derivatives $v_t = \nabla_x f_{\mathcal{B}_t}(x_t, y_t)$ and $w_t = \nabla_y f_{\mathcal{B}_t}(x_t, y_t)$;
6:     Given the mirror functions $\psi_t$ and $\phi_t$;
7:     $x_{t+1} = \arg\min_{x \in \mathcal{X}} \left\{ \langle v_t, x \rangle + \frac{1}{\gamma_t} D_{\psi_t}(x, x_t) + g(x) \right\}$;
8:     $y_{t+1} = y_t + \eta_t(\tilde{y}_{t+1} - y_t)$ where $\tilde{y}_{t+1} = \arg\max_{y \in \mathcal{Y}} \left\{ \langle w_t, y \rangle - \frac{1}{\lambda_t} D_{\phi_t}(y, y_t) - h(y) \right\}$;
9: **end for**
10: **Output:** $x_\zeta$ and $y_\zeta$ chosen uniformly random from $\{x_t, y_t\}_{t=1}^T$.
11: **Output:** (for theoretical) $x_\zeta$ and $y_\zeta$ chosen uniformly random from $\{x_t, y_t\}_{t=1}^T$.
12: **Output:** (for practical) $x_T$ and $y_T$.
---

keep it in the above subproblem (9) as the standard proximal descent algorithm [39]. Similarly, at step 8 of Algorithm 1, we use the mirror ascent to update $y$,

$$\tilde{y}_{t+1} = \arg\max_{y \in \mathcal{Y}} \left\{ \langle w_t, y \rangle - \frac{1}{\lambda_t} D_{\phi_t}(y, y_t) - h(y) \right\} \tag{10}$$

$$= \arg\max_{y \in \mathcal{Y}} \left\{ f(x_t, y_t) + \langle w_t, y - y_t \rangle - \frac{1}{\lambda_t} D_{\phi_t}(y, y_t) - h(y) \right\}. \tag{11}$$

In the above subproblem (11), the first two terms of its objective function is a linear function approximated the function $f(x, y)$ based on (stochastic) derivative estimator $w_t$, and the third term is a Bregman distance between $y$ and $y_t$ based on Bregman function $\phi_t$. Moreover, at the step 8 of Algorithm 1, we further use a momentum iteration to update $y$.

When Bregman functions $\psi_t(x) = \frac{1}{2}\|x\|^2$ and $\phi_t(y) = \frac{1}{2}\|y\|^2$ for all $t \geq 1$, we have $D_{\psi_t}(x, x_t) = \frac{1}{2}\|x - x_t\|^2$ and $D_{\phi_t}(y, y_t) = \frac{1}{2}\|y - y_t\|^2$. Under this case, Algorithm 1 will reduce the standard (stochastic) proximal gradient descent ascent algorithm. When Bregman functions $\psi_t(x) = \frac{1}{2}x^T H_t x$ and $\phi_t(y) = \frac{1}{2}y^T G_t y$ for all $t \geq 1$, we have $D_{\psi_t}(x, x_t) = \frac{1}{2}(x - x_t)^T H_t(x - x_t)$ and $D_{\phi_t}(y, y_t) = \frac{1}{2}(y - y_t)^T G_t(y - y_t)$, where $H_t \succeq \rho I_d$ and $G_t \succeq \rho I_p$. For example, given $\alpha \in (0, 1)$ and $\rho > 0$, we can generate the matrices $H_t$ and $G_t$ like as in Adam-type algorithms [21, 20], defined as

$$\tilde{v}_0 = 0, \ \tilde{v}_t = \alpha\tilde{v}_{t-1} + (1 - \alpha)\nabla_x f(x_t, y_t; \xi_t)^2, \quad H_t = \text{diag}(\sqrt{\tilde{v}_t} + \rho), \ t \geq 1 \tag{12}$$

$$\tilde{w}_0 = 0, \ \tilde{w}_t = \alpha\tilde{w}_{t-1} + (1 - \alpha)\nabla_y f(x_t, y_t; \xi_t)^2, \quad G_t = \text{diag}(\sqrt{\tilde{w}_t} + \rho), \ t \geq 1 \tag{13}$$

Under this case, our SMDA algorithm will reduce a novel adaptive gradient descent ascent algorithm.

In the problem (1), the functions $g(x)$ and $h(y)$ are generally nonsmooth, e.g., $g(x) = \nu_1\|x\|_1$ and $h(y) = \nu_2\|y\|_1$ with $\nu_1 > 0, \nu_2 > 0$. When Bregman functions $\psi_t(x) = \frac{1}{2}x^T H_t x$ and $\phi_t(y) = \frac{1}{2}y^T G_t y$ for all $t \geq 1$, and the matrices $H_t$ and $G_t$ are diagonal, e.g., generated from the above (12) and (13), we can use the soft thresholding operator $S(a, \lambda) = \text{sign}(a)\max(|a| - \lambda, 0) = \arg\min_z\{\frac{1}{2}(z - a)^2 + \lambda|z|\}$ to obtain the closed-form solutions of the following subproblems:

$$\min_{x \in \mathcal{X}} \left\{ \langle v_t, x \rangle + \frac{1}{2\gamma_t}(x - x_t)^T H_t(x - x_t) + \nu_1\|x\|_1 \right\}, \tag{14}$$

$$\max_{y \in \mathcal{Y}} \left\{ \langle w_t, y \rangle - \frac{1}{2\lambda_t}(y - y_t)^T G_t(y - y_t) - \nu_2\|y\|_1 \right\}, \tag{15}$$

where $H_t = \text{diag}(h_{1,t}, \cdots, h_{d,t})$ with $h_{i,t} > 0$ for $i \in [d]$, and $G_t = \text{diag}(g_{1,t}, \cdots, g_{p,t})$ with $g_{j,t} > 0$ for $j \in [p]$. Without loss of generality, let $\mathcal{X} = \mathbb{R}^d$ and $\mathcal{Y} = \mathbb{R}^p$, we have

$$S\left(x_{i,t} - \frac{\gamma_t}{h_{i,t}}v_{i,t}, \frac{\gamma_t\nu_1}{h_{i,t}}\right) = \arg\min_{x_i \in \mathbb{R}} \left\{ \langle v_{i,t}, x_i \rangle + \frac{h_{i,t}}{2\gamma_t}(x_i - x_{i,t})^2 + \nu_1|x_i| \right\}, \ i \in [d] \tag{16}$$

$$S\left(y_{j,t} + \frac{\lambda_t}{g_{j,t}}w_{j,t}, \frac{\lambda_t\nu_2}{g_{j,t}}\right) = \arg\max_{y_j \in \mathbb{R}} \left\{ \langle w_{j,t}, y_j \rangle - \frac{g_{j,t}}{2\lambda_t}(y_j - y_{j,t})^2 - \nu_2|y_j| \right\}, \ j \in [p]. \tag{17}$$

---

**Algorithm 2** Accelerated Stochastic Mirror Descent Ascent (VR-SMDA) Algorithm

---

1: **Input:** $T$, $q$, stepsizes $\{\gamma_t > 0, \lambda_t > 0, \eta_t \in (0, 1]\}_{t=1}^T$, mini-batch sizes $b$ and $b_1$;
2: **initialize:** $x_1 \in \mathcal{X}$ and $y_1 \in \mathcal{Y}$;
3: **for** $t = 1, 2, \ldots, T$ **do**
4:  **if** $\mod (t, q) = 0$ **then**
5:     Randomly generate mini-batch samples $\mathcal{B}_t = \{\xi_t^i\}_{i=1}^b$ with $|\mathcal{B}_t| = b$;
6:     Compute stochastic partial derivatives $v_t = \nabla_x f_{\mathcal{B}_t}(x_t, y_t)$ and $w_t = \nabla_y f_{\mathcal{B}_t}(x_t, y_t)$;
7:  **else**
8:     Randomly generate mini-batch samples $\mathcal{I}_t = \{\xi_t^i\}_{i=1}^{b_1}$ with $|\mathcal{I}_t| = b_1$;
9:     Compute stochastic partial derivatives

$$v_t = \nabla_x f_{\mathcal{I}_t}(x_t, y_t) - \nabla_x f_{\mathcal{I}_t}(x_{t-1}, y_{t-1}) + v_{t-1}, \tag{18}$$
$$w_t = \nabla_y f_{\mathcal{I}_t}(x_t, y_t) - \nabla_y f_{\mathcal{I}_t}(x_{t-1}, y_{t-1}) + w_{t-1}; \tag{19}$$

10:  **end if**
11:  Given the mirror functions $\psi_t$ and $\phi_t$;
12:  $x_{t+1} = \arg\min_{x \in \mathcal{X}} \left\{ \langle v_t, x \rangle + \frac{1}{\gamma_t} D_{\psi_t}(x, x_t) + g(x) \right\}$;
13:  $y_{t+1} = y_t + \eta_t(\tilde{y}_{t+1} - y_t)$ where $\tilde{y}_{t+1} = \arg\max_{y \in \mathcal{Y}} \left\{ \langle w_t, y \rangle - \frac{1}{\lambda_t} D_{\phi_t}(y, y_t) - h(y) \right\}$;
14: **end for**
15: **Output:** (for theoretical) $x_\zeta$ and $y_\zeta$ chosen uniformly random from $\{x_t, y_t\}_{t=1}^T$.
16: **Output:** (for practical) $x_T$ and $y_T$.

---

## 4.2 VR-SMDA Algorithm

In this subsection, we propose an accelerated stochastic mirror descent ascent (VR-SMDA) algorithm to solve the stochastic problem (1). Algorithm 2 describes the detailed algorithmic framework of the VR-SMDA method.

In Algorithm 1, we only draw a mini-batch samples $\mathcal{B}_t = \{\xi_t^i\}_{i=1}^b$ at each iteration. Clearly, the mini-batch samples will take large variances in our SMDA algorithm. Thus, we use the variance reduced technique of SPIDER in our VR-SMDA algorithm to accelerate it. Specifically, when $\mod (t, q) = 0$, we draw a relative large batch samples $\mathcal{B}_t = \{\xi_t^i\}_{i=1}^b$ to estimate our stochastic partial derivatives $v_t$ and $w_t$; when $\mod (t, q) \neq 0$, we only draw a mini-batch samples $\mathcal{I}_t = \{\xi_t^i\}_{i=1}^{b_1}$ ($b > b_1$) to estimate $v_t$ and $w_t$ in (18) and (19). Since samples $\mathcal{I}_t$ are independent to variables $\{x_t, x_{t-1}, y_t, y_{t-1}, v_{t-1}\}$, by using Assumption 2, we have

$$\mathbb{E}_{\mathcal{I}_t}[v_t] = \nabla_x f(x_t, y_t) - \nabla_x f(x_{t-1}, y_{t-1}) + v_{t-1} \neq \nabla_x f(x_t, y_t), \tag{20}$$
$$\mathbb{E}_{\mathcal{I}_t}[w_t] = \nabla_y f(x_t, y_t) - \nabla_y f(x_{t-1}, y_{t-1}) + w_{t-1} \neq \nabla_y f(x_t, y_t). \tag{21}$$

Thus, the partial derivative estimators $v_t$ and $w_t$ are biased. As in Algorithm 1, we also use the mirror descent iteration to update $x$, and use both the mirror ascent and momentum iterations to update $y$ in Algorithm 2.

## 5 Convergence Analysis

In this section, we study the convergence properties of our algorithms (*i.e.*, MDA, SMDA and VR-SMDA) under some mild conditions. All related proofs are provided in the supplementary materials. We first introduce a useful convergence metric $\|\mathcal{G}_t\|$ (or $\mathbb{E}\|\mathcal{G}_t\|$) as in [13, 25] to measure convergence properties of our algorithms. Given the parameters $x_t$ at $t$-th iteration by our algorithms, we define a gradient mapping [36, 13] as

$$\mathcal{G}_t = \frac{1}{\gamma_t}(x_t - x_{t+1}^+), \tag{22}$$

$$x_{t+1}^+ = \arg\min_{x \in \mathcal{X}} \left\{ \langle \nabla\Phi(x_t), x \rangle + \frac{1}{\gamma_t} D_{\psi_t}(x, x_t) + g(x) \right\}, \tag{23}$$

where $\Phi(x) = f(x, y^*(x)) - h(y^*(x)) = \max_{y \in \mathcal{Y}} \{f(x, y) - h(y)\}$. When $\mathcal{X} = \mathbb{R}^d$ and $g(x)$ is a constant, and $\psi_t(x) = \frac{1}{2}\|x\|^2$, we have $\mathcal{G}_t = \nabla\Phi(x_t) = \nabla_x f(x_t, y^*(x_t))$. Under this case, our convergence metric $\mathbb{E}\|\mathcal{G}_t\| = \mathbb{E}\|\nabla_x f(x_t, y^*(x_t))\|$ is a common convergence metric used in [27].

Since the objective function $f(x, y)$ is $\mu$-strongly concave over $y$, the standard (stochastic) proximal gradient ascent can easily obtain the global solution of the subproblem $\max_{y \in \mathcal{Y}}\{f(x, y) - h(y)\}$. Without loss of generalization, in our theoretical analysis, we give the mirror functions $\phi_t(y) = \frac{1}{2}\|y\|^2$ for all $t \geq 1$, and all mirror functions $\{\psi_t(x)\}_{t=1}^T$ are $\rho$-strong convex. Here, the constant $\rho$ can be seen as a lower bound of the strong convexity of all functions $\{\psi_t(x)\}_{t=1}^T$ as in [25].

## 5.1 Convergence Analysis of the SMDA and MDA Algorithms

In the subsection, we provide the convergence properties of our SMDA and MDA algorithms.

**Theorem 1.** *Suppose the sequence $\{x_t, y_t\}_{t=1}^T$ be generated from Algorithm 1 using stochastic partial derivatives (i.e., SMDA algorithm). Let $0 < \eta = \eta_t \leq 1$, $0 < \gamma = \gamma_t \leq \min(\frac{3\rho}{4L}, \frac{9\eta\rho\mu\lambda}{800\kappa^2}, \frac{2\eta\mu\rho\lambda}{25L_f^2})$ and $0 < \lambda = \lambda_t \leq \frac{1}{6L_f}$, we have*

$$\frac{1}{T}\sum_{t=1}^T \mathbb{E}\|\mathcal{G}_t\| \leq \frac{4\sqrt{2(\tilde{F}(x_1) - F^*)}}{\sqrt{3T\gamma\rho}} + \frac{4\sqrt{2}\Delta_1}{\sqrt{3T\gamma\rho}} + \frac{10\sigma}{\sqrt{3b\rho}} + \frac{20\sigma\sqrt{\eta\lambda}}{3\sqrt{\gamma\rho\mu b}}, \tag{24}$$

*where $\kappa = L_f/\mu$, $L = L_f(1 + \kappa)$, $\tilde{F}(x) = \Phi(x) + g(x)$ and $\Delta_1 = \|y_1 - y^*(x_1)\|$.*

**Remark 1.** *Without loss of generality, let $L_f \geq \frac{1}{\mu}$. Given $0 < \eta \leq 1$, $\lambda = O(\frac{1}{L_f})$, $\gamma = \min(\frac{3\rho}{4L}, \frac{9\eta\rho\mu\lambda}{800\kappa^2}, \frac{2\eta\mu\rho\lambda}{25L_f^2})$ and $\rho = O(L_f^\nu)$ ($\nu \geq 0$), we have $\gamma = O(\kappa^{\nu-3})$ and $\gamma\rho = O(\kappa^{2\nu-3})$. Thus, our SMDA algorithm has a convergence rate of $O\left(\sqrt{\frac{\kappa^{3-2\nu}}{T}} + \sqrt{\frac{\kappa^{-2\nu}}{b}} + \sqrt{\frac{\kappa^{3-2\nu}}{b}}\right)$. When let $\nu = \frac{1}{2}$, $b = T/\kappa$ and $\sqrt{\frac{\kappa^2}{T}} = \epsilon/3$, we have $T = O(\kappa^2\epsilon^{-2})$ and $b = O(\kappa\epsilon^{-2})$. Since our SMDA algorithm requires $2b$ stochastic gradient evaluations to estimate the stochastic partial directives $v_t$ and $w_t$ at each iteration, and needs $T$ iterations, it has a gradient complexity of $2bT = O(\kappa^3\epsilon^{-4})$ for finding an $\epsilon$-stationary point, the same complexity in [4]. When let $\nu = 4/3$, $b = T/\kappa^{1/3}$ and $\sqrt{\frac{\kappa^{1/3}}{T}} = \epsilon/3$, we have $T = O(\kappa^{1/3}\epsilon^{-2})$ and $b = O(\epsilon^{-2})$. Thus, our SMDA algorithm has a near optimal gradient complexity of $2bT = O(\kappa^{1/3}\epsilon^{-4})$, which matches a gradient complexity lower bound given in [24] for solving the problem* (1) *without the nonsmooth regularization terms.*

**Theorem 2.** *Suppose the sequence $\{x_t, y_t\}_{t=1}^T$ be generated from Algorithm 1 using the deterministic partial derivatives (i.e., MDA algorithm). Let $0 < \eta = \eta_t \leq 1$, $0 < \gamma = \gamma_t \leq \min(\frac{3\rho}{4L}, \frac{9\eta\rho\mu\lambda}{800\kappa^2}, \frac{2\eta\mu\rho\lambda}{25L_f^2})$ and $0 < \lambda = \lambda_t \leq \frac{1}{6L_f}$, we have*

$$\frac{1}{T}\sum_{t=1}^T \|\mathcal{G}_t\| \leq \frac{4\sqrt{2(\tilde{F}(x_1) - F^*)}}{\sqrt{3T\gamma\rho}} + \frac{4\sqrt{2}\Delta_1}{\sqrt{3T\gamma\rho}}, \tag{25}$$

*where $\kappa = L_f/\mu$, $L = L_f(1 + \kappa)$, $\tilde{F}(x) = \Phi(x) + g(x)$ and $\Delta_1 = \|y_1 - y^*(x_1)\|$.*

**Remark 2.** *Without loss of generality, let $L_f \geq \frac{1}{\mu}$. Given $0 < \eta \leq 1$, $\lambda = O(\frac{1}{L_f})$, $\gamma = \min(\frac{3\rho}{4L}, \frac{9\eta\rho\mu\lambda}{800\kappa^2}, \frac{2\eta\mu\rho\lambda}{25L_f^2})$ and $\rho = O(L_f^{(\frac{1}{2}+\nu)})$ ($\nu \geq 0$), we have $\frac{1}{\gamma\rho} = O(\kappa^{(2-2\nu)})$. Since our MDA algorithm requires 2 gradient evaluations at each iteration, and needs $T$ iterations, it has a gradient complexity of $2T = O(\kappa^{(2-2\nu)}\epsilon^{-2})$ for finding an $\epsilon$-stationary point. When let $\nu = 0$, our MDA algorithm has a gradient complexity of $2T = O(\kappa^2\epsilon^{-2})$, the same complexity in [4]. When let $\nu = 1/2$, our MDA algorithm has a lower gradient complexity of $T = O(\kappa\epsilon^{-2})$ than the complexity in [4, 2]. When let $\nu = 3/4$, our MDA algorithm has a near optimal gradient complexity of $T = O(\sqrt{\kappa}\epsilon^{-2})$, which is the same complexity in [28] for solving the problem* (1) *without the nonsmooth regularization terms.*

## 5.2 Convergence Analysis of the VR-SMDA Algorithm

In the subsection, we provide the convergence properties of the VR-SMDA algorithm.

**Theorem 3.** *Suppose the sequence $\{x_t, y_t\}_{t=1}^T$ be generated from Algorithm 2. Let $b_1 = q$, $0 < \eta = \eta_t \leq 1$, $0 < \gamma = \gamma_t \leq \min(\frac{3\rho}{4L}, \frac{\eta\mu\lambda\rho}{38L_f^2}, \frac{3\rho}{19L_f^2\eta}, \frac{\rho\eta}{8}, \frac{9\rho\eta\mu\lambda}{400\kappa^2})$ and $0 < \lambda = \lambda_t \leq \min(\frac{1}{6L_f}, \frac{9\mu}{100\eta^2L_f^2})$,*

*we have*

$$\frac{1}{T}\sum_{t=1}^{T}\mathbb{E}\|\mathcal{G}_t\| \leq \frac{4\sqrt{2(\tilde{F}(x_1)-F^*)}}{\sqrt{3T\gamma\rho}} + \frac{4\sqrt{2}\Delta_1}{\sqrt{3T\gamma\rho}} + \frac{2\sqrt{2}\sigma}{\sqrt{\gamma\rho\eta b}L_f}, \tag{26}$$

*where $\kappa = L_f/\mu$, $L = L_f(1+\kappa)$, $\tilde{F}(x) = \Phi(x) + g(x)$ and $\Delta_1 = \|y_1 - y^*(x_1)\|$.*

**Remark 3.** *Without loss of generality, let $L_f \geq \frac{1}{\mu}$. Given $0 < \eta \leq 1$, $\lambda = O(\frac{1}{\kappa L_f})$, $\gamma = \min(\frac{3\rho}{4L}, \frac{\eta\mu\lambda\rho}{38L_f^2}, \frac{3\rho}{19L_f^2\eta}, \frac{\rho\eta}{8}, \frac{9\rho\eta\mu\lambda}{400\kappa^2})$ and $\rho = O(L_f^{1+\nu})$ $(\nu \geq 0)$, we have $\frac{1}{\gamma\rho} = O(\kappa^{2-2\nu})$. Thus, our VR-SMDA algorithm has a convergence rate of $O(\sqrt{\frac{\kappa^{(2-2\nu)}}{T}} + \sqrt{\frac{\kappa^{(1-2\nu)}}{b}})$. When let $\nu = 0$, $b = T/\kappa$ and $\sqrt{\frac{\kappa^2}{T}} = \epsilon/2$, we have $T = O(\kappa^2\epsilon^{-2})$. Further let $b_1 = q = O(\kappa\epsilon^{-1})$ and $b = O(\kappa\epsilon^{-2})$. Since our VR-SMDA algorithm requires $2b$ stochastic gradient evaluations to estimate the stochastic directives $v_t$ and $w_t$ at each iteration when $\mod(t,q) = 0$, otherwise needs $4b_1$ stochastic gradient evaluations, and need $T$ iterations, it has a gradient complexity of $4b_1 T + 2bT/q = O(\kappa^3\epsilon^{-3})$ for finding an $\epsilon$-stationary point of the problem* (1)*, which is the same complexity in [30] for solving the problem* (1) *without the nonsmooth regularization terms.*

**Remark 4.** *The above optimal gradient complexities are obtained when given $\rho = O(L_f^{\nu'})$ $(\nu' > 0)$, where $L_f$ is the smooth parameter of objective function $f(x,y)$. Although in the objective function $f(x,y)$, $L_f$ may be large, we can easily change the original objective function $f(x,y)$ to a new function $\hat{f}(x,y) = \tau f(x,y)$, $0 < \tau < 1$. Since $\nabla\hat{f}(x,y) = \tau\nabla f(x,y)$, the gradient of function $\hat{f}(x,y)$ is $\hat{L}$-Lipschitz continuous ($\hat{L} = \tau L_f$). Thus, we can choose a suitable hyper-parameter $\tau$ to let this new objective function $\hat{f}(x,y)$ satisfy the condition $\rho = O(\hat{L})$.*

## 6 Numerical Experiments

In this section, we perform two tasks (i.e., fair classifier and robust neural network training) to validate efficiency of our algorithms. Specifically, we conduct these tasks on the Fashion-MNIST dataset as in [38] as well MNIST dataset and CIFAR-10 dataset. Fashion-MNIST dataset and MNIST dataset consist of $28 \times 28$ arrays of grayscale pixel images classified into 10 categories, and includes $60,000$ training images and $10,000$ testing images. CIFAR-10 dataset includes $60,000$ $32 \times 32$ colour images ($50,000$ training images and $10,000$ testing images). In the experiment, we compare our algorithms (MDA, SMDA and VR-SMDA) with the existing proximal gradient descent ascent algorithms (MAPGDA [2], PAGDA [4] and PASGDA[4] ) for solving these nonsmooth nonconvex minimax problems. Note that both HiBSA algorithm of [29] and Proximal-GDA algorithm of [8] only are a non-accelerated version of MAPGDA algorithm [2], so we omit them in the comparison methods. The experiments are run on CPU machines with 2.3 GHz Intel Core i9 as well as NVIDIA Tesla P40 GPU.

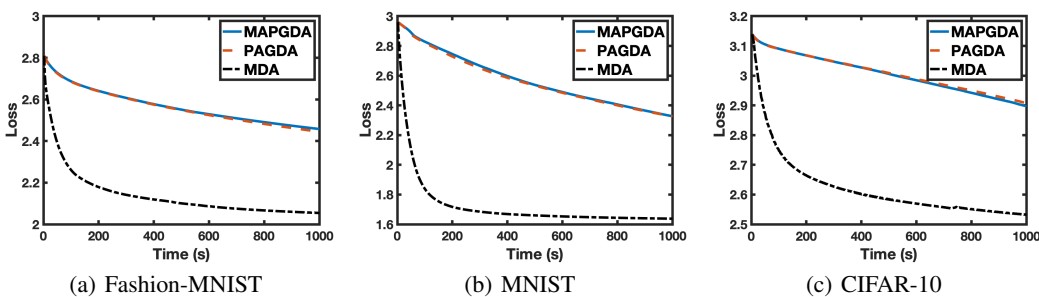

(a) Fashion-MNIST       (b) MNIST       (c) CIFAR-10

Figure 1: Results of different deterministic methods on the fair classifier task.

### 6.1 Fair Classifier

The first task is to train a fair classifier to minimize the maximum loss over categories. Here, we use a nonconvex Convolutional Neural Network (CNN) model as classifier. Similar to [38], we limit our experiment to the three categories. To be precise, the Fashion-MNIST dataset is limited to T-shirt/top,

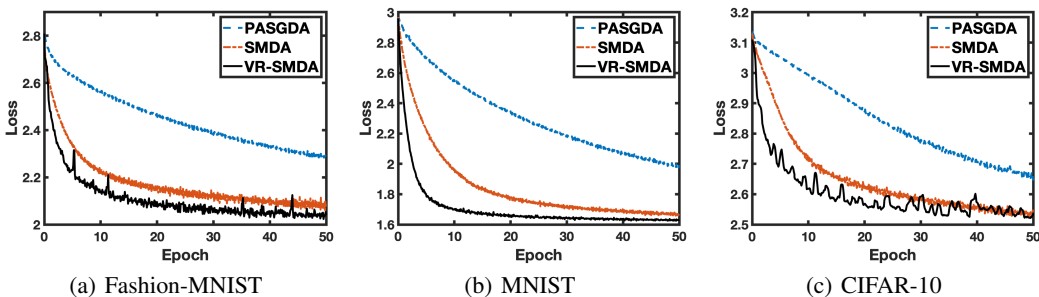

|          (a) Fashion-MNIST          |          (b) MNIST          |          (c) CIFAR-10          |

Figure 2: Results of different stochastic methods on the fair classifier task.

Coat and Shirt categories. The MNIST dataset is limited to digital numbers $\{0, 2, 3\}$, and CIFAR10 dataset is limited to airplane, automobile and bird. Specifically, we will solve the minimax problem:

$$\min_{w} \max_{y \in \mathcal{Y}} \left\{ \sum_{i=1}^{3} y_i \mathcal{L}_i(w) + g(w) - h(y) \right\}, \quad \text{s.t. } \mathcal{Y} = \{y \in \mathbb{R}^3 \mid y_i \geq 0, \sum_{i=1}^{3} y_i = 1\}, \quad (27)$$

where $w$ denotes the parameters in CNN model, and $\mathcal{L}_1$, $\mathcal{L}_2$ and $\mathcal{L}_3$ are the loss functions corresponding to the samples in three different categories. Here we let $g(w) = \nu_1 \|w\|_1$ and $h(y) = \nu_2 \|y\|_2^2$, where $\nu_1 > 0$ and $\nu_2 > 0$. Clearly the inner maximization problem is strongly concave, and the outer minimization problem is nonconvex nonsmooth. Thus our theory can be applied.

In the experiment, we let $\nu_1 = 0.001$ and $\nu_2 = 0.1$ in the above problem (27). For fair comparison, we use the same step size for all methods. Specifically, step-size for $w$ is 0.001 and step-size for $y$ is 0.00001. We apply xavier normal initialization to CNN layer. In our algorithms, we choose the mirror functions $\psi_t(w) = \frac{1}{2} w^T H_t w$ and $\phi_t(y) = \frac{1}{2} y^T G_t y$ for all $t \geq 1$, where $H_t$ and $G_t$ are generated from (12) and (13) respectively, given $\alpha = 0.1$ and $\rho = 0.00005$. We set $\eta = \eta_t = 1$ in our algorithms. We run all deterministic algorithms for 1000 seconds and all stochastic algorithms for 50 epochs. Then we record the loss value. For stochastic methods, batch sizes of PASGDA and SMDA are 3000. For our VR-SMDA, we set the large batch size $b = 60000$ and the mini-batch size $b_1 = q = 3000$.

Figure 1 shows the loss vs time of different deterministic methods. Figure 2 plots the loss vs epoch of different stochastic methods. From these results, we can find that our algorithms consistently outperform the other algorithms with a great margin. The main reason is that our algorithms use the preconditioned (adaptive) matrices $H_t$ and $G_t$ in updating $x$ and $y$, respectively.

## 6.2 Robust Neural Network Training

The second task is to train robust Neural Networks (NNs). Although the NNs have been widely used in many applications such as image classification, they are vulnerable to adversarial attacks such as Fast Gradient Sign Method (FGSM) [15] and Projected Gradient Descent (PGD) attack [22]. In other word, a small perturbation in the input of NN can significantly change its output. Thus, we try to train a robust NN against these adversarial attacks, which generally reformulate this robust training into the following minimax problem:

$$\min_{w} \sum_{i=1}^{n} \max_{y_i \in \mathcal{Y}} \mathcal{L}\big(f(a_i + y_i; w), b_i\big), \quad \mathcal{Y} = \{y_i \in \mathbb{R}^d \mid \|y_i\|_\infty \leq \varepsilon, \ i \in [n]\} \quad (28)$$

where $(a_i, b_i)$ denotes the $i$-th data point, and $w$ is the parameter of NN, and $y_i \in \mathbb{R}^d$ denotes is the perturbation added to the $i$-th data point. Following [38], we approximate the inner maximization problem of the above minimax problem (28) with the following finite max problem

$$\min_{w} \sum_{i=1}^{n} \max \left\{ \mathcal{L}\big(f(\hat{a}_{i,0}(w); w), b_i\big), \cdots, \mathcal{L}\big(f(\hat{a}_{i,9}(w); w), b_i\big) \right\}, \quad (29)$$

where $\hat{a}_{i,j}(w)$ is the result of a targeted attack on data point $a_i$ that is changed the output of NN to label $j$. Following [38], we can obtain $\hat{a}_{i,j}(w)$ by using the following procedure: In the last layer of the NN architecture for learning classification on MNIST (Fashion-MNIST) we have 10 different

neurons, each corresponding with one category of classification. For any sample $(a_i, b_i)$ in the dataset and starting from $a_{i,j}^0 = a_i$ for any $j = 0, 1, \cdots, 9$, we run projected gradient ascent to obtain the following chain of points:

$$a_{i,j}^{k+1} = \text{Proj}_\mathcal{Y}\left[a_{i,j}^k + \mu\nabla_a\left(Z_j(a_{i,j}^k, w) - Z_{b_i}(a_{i,j}^k, w)\right)\right], \ k = 0, 1, \cdots, K-1 \quad (30)$$

where $\mu > 0$ is a stepsize, and $Z_j$ is the network logit before softmax corresponding to label $j$. Finally, we can set $\hat{a}_{i,j}(w) = a_{i,j}^K$ in the above minimax problem (29).

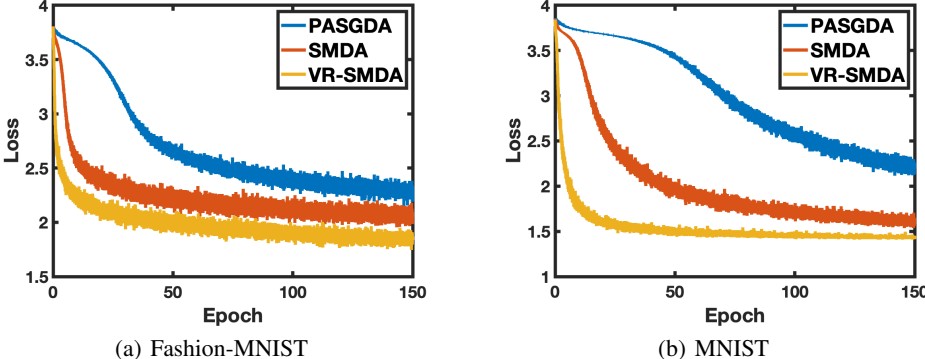

(a) Fashion-MNIST                         (b) MNIST

Figure 3: Results of different stochastic methods on the robust NN training task at Fashion-MNIST and MNIST datasets.

Next, we can replace the above problem (29) with the following nonconvex nonsmooth problem:

$$\min_w \sum_{i=1}^n \max_{u \in U} \left\{ \sum_{j=0}^9 u_j \, \mathcal{L}\left(f(a_{i,j}^K; w), b_i\right) + \nu_1\|w\|_1 - \nu_2\|u\|_2^2 \right\}, \quad (31)$$

$$\text{s.t. } U = \left\{ u \in \mathbb{R}^{10} \mid u_j \geq 0, \ \sum_{j=0}^9 u_j = 1 \right\},$$

where $\nu_1 > 0$ and $\nu_2 > 0$. Clearly, the inner maximization problem in (31) is strongly-concave, and its outer minimization problem is nonconvex and nonsmooth.

In the experiment, we set $\nu_1 = 0.0001$ and $\nu_2 = 0.1$ in the above problem (31). In the above problem (30), we set $K = 5$. For fair comparison, we use the same step size for all methods. Specifically, step-size for $w$ is 0.0005 and step-size for $u$ is 0.00001. We set $\eta = \eta_t = 1$ in our algorithms. For our algorithms, we choose the mirror functions $\psi_t(w) = \frac{1}{2}w^T H_t w$ and $\phi_t(u) = \frac{1}{2}u^T G_t u$ for all $t \geq 1$, where $H_t$ and $G_t$ are generated from (12) and (13) respectively, given $\alpha = 0.1$ and $\rho = 0.0005$. Here we only conduct experiments with stochastic methods, and batch-sizes of PASGDA and SMDA are 600. For our VR-SMDA, we set $b = 1200$ and $b_1 = q = 600$. Following [38], we set $\varepsilon = 0.4$ in the above problem (28). Figure 3 shows the loss vs epoch of different stochastic methods. From these results, we can find that our algorithms outperform the other algorithms, and the VR-SMDA consistently outperforms the SMDA.

## 7 Conclusions

In the paper, we proposed a class of novel adaptive mirror descent ascent methods to solve the nonconvex-strongly-concave minimax optimization problems with nonsmooth regularization terms. Moreover, we provided a useful convergence analysis framework for our methods. Some experimental results on fair classifier and robust neural network training tasks verify that our new algorithms consistently outperform the related algorithms.

## Acknowledgments and Disclosure of Funding

This work was partially supported by NSF IIS 1845666, 1852606, 1838627, 1837956, 1956002, OIA 2040588.

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
