# A  Supplementary Materials

In this section, we provide the detailed convergence analysis of our algorithms. We first gives some useful lemmas.

**Lemma 1.** *( Proposition 1 of [8] ) Let $y^*(x) = \arg\max_{y \in \mathcal{Y}}\{f(x,y) - h(y)\}$ and $\Phi(x) = \max_{y \in \mathcal{Y}}\{f(x,y) - h(y)\}$. Under the above assumptions, the mapping $y^*(x)$ and the function $\Phi(x)$ satisfy*

   *1)  Mapping $y^*(x)$ is $\kappa$-Lipschitz continuous;*

   *2)  Function $\Phi(x)$ is $L_f(1+\kappa)$-smooth with $\nabla\Phi(x) = \nabla_x f(x, y^*(x))$,*

*where $\kappa = L_f/\mu$ denotes the condition number of function $f(x,y)$.*

**Lemma 2.** *[36] Let $f(x)$ is a convex function and $\mathcal{X}$ is a convex set. $x^* \in \mathcal{X}$ is the solution of the constrained problem $\min_{x \in \mathcal{X}} f(x)$, if*

$$\langle \nabla f(x^*), x - x^* \rangle > 0, \ \forall x \in \mathcal{X}. \tag{32}$$

*where $\nabla f(x)$ denote gradient of the function $f(x)$.*

**Lemma 3.** *For independent random variables $\{\xi_i\}_{i=1}^n$ with zero mean, we have $\mathbb{E}\|\frac{1}{n}\sum_{i=1}^n \xi_i\|^2 = \frac{1}{n}\mathbb{E}\|\xi_i\|^2$ for any $i \in [n]$.*

## A.1  Convergence Analysis of the SMDA and MDA Algorithms

In the subsection, we study the convergence properties of the SMDA and MDA algorithms for solving the minimax problem (1). We first provide some useful lemmas.

**Lemma 4.** *(Lemma 1 in [13]) Let $x_{t+1} = \arg\min_{x \in \mathcal{X}}\left\{\langle v_t, x\rangle + \frac{1}{\gamma_t}D_{\psi_t}(x, x_t) + g(x)\right\}$ and $\tilde{\mathcal{G}}_t = \frac{1}{\gamma_t}(x_t - x_{t+1})$, we have, for all $t \geq 1$*

$$\langle v_t, \tilde{\mathcal{G}}_t \rangle \geq \rho\|\tilde{\mathcal{G}}_t\|^2 + \frac{1}{\gamma_t}\big(g(x_{t+1}) - g(x_t)\big), \tag{33}$$

*where $\rho > 0$ depends on $\rho$-strongly convex function $\psi_t(x)$.*

**Lemma 5.** *Let $x_{t+1}$ be generated from Algorithm 1 or 2, and define $x_{t+1}^+ = \arg\min_{x \in \mathcal{X}}\{\langle \nabla\Phi(x_t), x\rangle + \frac{1}{\gamma_t}D_{\psi_t}(x, x_t) + g(x)\}$, and let $\mathcal{G}_t = \frac{1}{\gamma_t}(x_t - x_{t+1}^+)$, $\tilde{\mathcal{G}}_t = \frac{1}{\gamma_t}(x_t - x_{t+1})$, we have*

$$\|\mathcal{G}_t - \tilde{\mathcal{G}}_t\| \leq \frac{1}{\rho}\|\nabla\Phi(x_t) - v_t\|, \tag{34}$$

*where $\Phi(x_t) = \max_{y \in \mathcal{Y}}\{f(x_t, y) - h(y)\}$ and $\rho > 0$ depends on $\rho$-strongly convex function $\psi_t(x)$.*

*Proof.* Since $x_{t+1} = \arg\min_{x \in \mathcal{X}}\left\{\langle v_t, x\rangle + \frac{1}{\gamma_t}D_{\psi_t}(x, x_t) + g(x)\right\}$ and $x_{t+1}^+ = \arg\min_{x \in \mathcal{X}}\left\{\langle \nabla\Phi(x_t), x\rangle + \frac{1}{\gamma_t}D_{\psi_t}(x, x_t) + g(x)\right\}$, by Lemma 2, we have, for all $x \in \mathcal{X}$

$$\langle v_t + \nabla g(x_{t+1}) + \frac{1}{\gamma_t}(\nabla\psi_t(x_{t+1}) - \nabla\psi_t(x_t)), x - x_{t+1}\rangle \geq 0, \tag{35}$$

$$\langle \nabla\Phi(x_t) + \nabla g(x_{t+1}^+) + \frac{1}{\gamma_t}(\nabla\psi_t(x_{t+1}^+) - \nabla\psi_t(x_t)), x - x_{t+1}^+\rangle \geq 0, \tag{36}$$

where $\nabla g(x_{t+1}) \in \partial g(x_{t+1})$. Taking $x = x_{t+1}^+$ in the inequality (35) and $x = x_{t+1}$ in the inequality (36), by the convexity of $g(x)$, we have

$$\langle v_t, x_{t+1}^+ - x_{t+1}\rangle \geq \langle \nabla g(x_{t+1}), x_{t+1} - x_{t+1}^+\rangle + \frac{1}{\gamma_t}\langle\nabla\psi_t(x_{t+1}) - \nabla\psi_t(x_t), x_{t+1} - x_{t+1}^+\rangle, \tag{37}$$

$$\geq g(x_{t+1}) - g(x_{t+1}^+) + \frac{1}{\gamma_t}\langle\nabla\psi_t(x_{t+1}) - \nabla\psi_t(x_t), x_{t+1} - x_{t+1}^+\rangle$$

$$\langle \nabla\Phi(x_t), x_{t+1} - x_{t+1}^+\rangle \geq \langle \nabla g(x_{t+1}^+), x_{t+1}^+ - x_{t+1}\rangle + \frac{1}{\gamma_t}\langle\nabla\psi_t(x_{t+1}^+) - \nabla\psi_t(x_t), x_{t+1}^+ - x_{t+1}\rangle, \tag{38}$$

$$\geq g(x_{t+1}^+) - g(x_{t+1}) + \frac{1}{\gamma_t}\langle\nabla\psi_t(x_{t+1}^+) - \nabla\psi_t(x_t), x_{t+1}^+ - x_{t+1}\rangle$$

Summing up the above inequalities (37) and (38), we obtain

$$\langle \nabla\Phi(x_t) - v_t, x_{t+1} - x_{t+1}^+\rangle \geq \frac{1}{\gamma_t}\langle\nabla\psi_t(x_{t+1}^+) - \nabla\psi_t(x_{t+1}), x_{t+1}^+ - x_{t+1}\rangle$$

$$\geq \frac{\rho}{\gamma_t}\|x_{t+1}^+ - x_{t+1}\|^2, \tag{39}$$

where the last inequality is due to the $\rho$-strongly convex function $\psi_t(x)$.

Since $\|\nabla\Phi(x_t) - v_t\|\|x_{t+1} - x_{t+1}^+\| \geq \langle\nabla\Phi(x_t) - v_t, x_{t+1} - x_{t+1}^+\rangle$ and $\|\mathcal{G}_t - \tilde{\mathcal{G}}_t\| = \|\frac{1}{\gamma_t}(x_t - x_{t+1}^+) - \frac{1}{\gamma_t}(x_t - x_{t+1})\| = \frac{1}{\gamma_t}\|x_{t+1}^+ - x_{t+1}\|$, we have

$$\|\nabla\Phi(x_t) - v_t\| \geq \rho\|\mathcal{G}_t - \tilde{\mathcal{G}}_t\|. \tag{40}$$

$\square$

**Lemma 6.** *Suppose the sequence $\{x_t, y_t\}_{t=1}^T$ be generated from Algorithm 1 or 2. Under the above assumptions, and let $0 < \eta_t \leq 1$, $\lambda = \lambda_t$ and $0 < \lambda \leq \frac{1}{6L_f}$, we have*

$$\|y_{t+1} - y^*(x_{t+1})\|^2 - \|y_t - y^*(x_t)\|^2 \leq -\frac{\eta_t\mu\lambda}{4}\|y_t - y^*(x_t)\|^2 - \frac{3\eta_t}{4}\|\tilde{y}_{t+1} - y_t\|^2$$
$$+ \frac{25\eta_t\lambda}{6\mu}\|\nabla_y f(x_t, y_t) - w_t\|^2 + \frac{25\kappa^2}{6\eta_t\mu\lambda}\|x_t - x_{t+1}\|^2, \tag{41}$$

*where $\kappa = L_f/\mu$.*

*Proof.* This proof mainly follows the proof of Lemma 28 in [18]. According to Assumption 3, i.e., the function $f(x, y)$ is $\mu$-strongly concave w.r.t $y$, we have

$$f(x_t, y) \leq f(x_t, y_t) + \langle\nabla_y f(x_t, y_t), y - y_t\rangle - \frac{\mu}{2}\|y - y_t\|^2$$
$$= f(x_t, y_t) + \langle w_t, y - \tilde{y}_{t+1}\rangle + \langle\nabla_y f(x_t, y_t) - w_t, y - \tilde{y}_{t+1}\rangle$$
$$+ \langle\nabla_y f(x_t, y_t), \tilde{y}_{t+1} - y_t\rangle - \frac{\mu}{2}\|y - y_t\|^2. \tag{42}$$

According to Assumption 1, i.e., the function $f(x, y)$ is $L_f$-smooth, we have

$$-\frac{L_f}{2}\|\tilde{y}_{t+1} - y_t\|^2 \leq f(x_t, \tilde{y}_{t+1}) - f(x_t, y_t) - \langle\nabla_y f(x_t, y_t), \tilde{y}_{t+1} - y_t\rangle. \tag{43}$$

Summing up the about inequalities (42) with (43), we have

$$f(x_t, y) \leq f(x_t, \tilde{y}_{t+1}) + \langle w_t, y - \tilde{y}_{t+1}\rangle + \langle\nabla_y f(x_t, y_t) - w_t, y - \tilde{y}_{t+1}\rangle$$
$$- \frac{\mu}{2}\|y - y_t\|^2 + \frac{L_f}{2}\|\tilde{y}_{t+1} - y_t\|^2. \tag{44}$$

Given the mirror function $\phi_t(y) = \frac{1}{2}\|y\|$ and $\lambda = \lambda_t$ for all $t \geq 1$, at the step 8 of Algorithm 1 ( at step 13 of Algorithm 2 ), we have

$$\tilde{y}_{t+1} = \arg\max_{y\in\mathcal{Y}}\left\{\langle w_t, y\rangle - \frac{1}{2\lambda}\|y - y_t\|^2 - h(y)\right\}. \tag{45}$$

By using Lemma 2, we have

$$\langle -w_t + \nabla h(\tilde{y}_{t+1}) + \frac{1}{\lambda}(\tilde{y}_{t+1} - y_t), y - \tilde{y}_{t+1}\rangle \geq 0, \ \forall y \in \mathcal{Y} \tag{46}$$

where $\nabla h(\tilde{y}_{t+1}) \in \partial h(\tilde{y}_{t+1})$. Then we obtain

$$\langle w_t, y - \tilde{y}_{t+1}\rangle \leq \frac{1}{\lambda}\langle\tilde{y}_{t+1} - y_t, y - \tilde{y}_{t+1}\rangle + \langle\nabla h(\tilde{y}_{t+1}), y - \tilde{y}_{t+1}\rangle$$
$$\leq \frac{1}{\lambda}\langle\tilde{y}_{t+1} - y_t, y - \tilde{y}_{t+1}\rangle + h(y) - h(\tilde{y}_{t+1})$$
$$= -\frac{1}{\lambda}\|\tilde{y}_{t+1} - y_t\|^2 + \frac{1}{\lambda}\langle\tilde{y}_{t+1} - y_t, y - y_t\rangle + h(y) - h(\tilde{y}_{t+1}). \tag{47}$$

where the second inequality holds by the convexity of function $h(y)$.

By pugging the inequalities (47) into (44), we have

$$f(x_t, y) - h(y) \leq f(x_t, \tilde{y}_{t+1}) - h(\tilde{y}_{t+1}) + \frac{1}{\lambda}\langle\tilde{y}_{t+1} - y_t, y - y_t\rangle + \langle\nabla_y f(x_t, y_t) - w_t, y - \tilde{y}_{t+1}\rangle$$
$$- \frac{1}{\lambda}\|\tilde{y}_{t+1} - y_t\|^2 - \frac{\mu}{2}\|y - y_t\|^2 + \frac{L_f}{2}\|\tilde{y}_{t+1} - y_t\|^2. \tag{48}$$

Let $y = y^*(x_t)$ and we obtain

$$f(x_t, y^*(x_t)) - h(y^*(x_t)) \leq f(x_t, \tilde{y}_{t+1}) - h(\tilde{y}_{t+1}) + \frac{1}{\lambda}\langle\tilde{y}_{t+1} - y_t, y^*(x_t) - y_t\rangle - (\frac{1}{\lambda} - \frac{L_f}{2})\|\tilde{y}_{t+1} - y_t\|^2$$
$$+ \langle\nabla_y f(x_t, y_t) - w_t, y^*(x_t) - \tilde{y}_{t+1}\rangle - \frac{\mu}{2}\|y^*(x_t) - y_t\|^2. \tag{49}$$

Due to the concavity of $f(\cdot, y) - h(y)$ and $y^*(x_t) = \arg\max_{y \in \mathcal{Y}}\{f(x_t, y) - h(y)\}$, we have $f(x_t, y^*(x_t)) - h(y^*(x_t)) \geq f(x_t, \tilde{y}_{t+1}) - h(\tilde{y}_{t+1})$. Thus, we obtain

$$0 \leq \frac{1}{\lambda}\langle \tilde{y}_{t+1} - y_t, y^*(x_t) - y_t\rangle + \langle \nabla_y f(x_t, y_t) - w_t, y^*(x_t) - \tilde{y}_{t+1}\rangle$$
$$- (\frac{1}{\lambda} - \frac{L_f}{2})\|\tilde{y}_{t+1} - y_t\|^2 - \frac{\mu}{2}\|y^*(x_t) - y_t\|^2. \tag{50}$$

By $y_{t+1} = y_t + \eta_t(\tilde{y}_{t+1} - y_t)$, we have

$$\|y_{t+1} - y^*(x_t)\|^2 = \|y_t + \eta_t(\tilde{y}_{t+1} - y_t) - y^*(x_t)\|^2$$
$$= \|y_t - y^*(x_t)\|^2 + 2\eta_t\langle \tilde{y}_{t+1} - y_t, y_t - y^*(x_t)\rangle + \eta_t^2\|\tilde{y}_{t+1} - y_t\|^2. \tag{51}$$

Then we obtain

$$\langle \tilde{y}_{t+1} - y_t, y^*(x_t) - y_t\rangle \leq \frac{1}{2\eta_t}\|y_t - y^*(x_t)\|^2 + \frac{\eta_t}{2}\|\tilde{y}_{t+1} - y_t\|^2 - \frac{1}{2\eta_t}\|y_{t+1} - y^*(x_t)\|^2. \tag{52}$$

Consider the upper bound of the term $\langle \nabla_y f(x_t, y_t) - w_t, y^*(x_t) - \tilde{y}_{t+1}\rangle$, we have

$$\langle \nabla_y f(x_t, y_t) - w_t, y^*(x_t) - \tilde{y}_{t+1}\rangle$$
$$= \langle \nabla_y f(x_t, y_t) - w_t, y^*(x_t) - y_t\rangle + \langle \nabla_y f(x_t, y_t) - w_t, y_t - \tilde{y}_{t+1}\rangle$$
$$\leq \frac{1}{\mu}\|\nabla_y f(x_t, y_t) - w_t\|^2 + \frac{\mu}{4}\|y^*(x_t) - y_t\|^2 + \frac{1}{\mu}\|\nabla_y f(x_t, y_t) - w_t\|^2 + \frac{\mu}{4}\|y_t - \tilde{y}_{t+1}\|^2$$
$$= \frac{2}{\mu}\|\nabla_y f(x_t, y_t) - w_t\|^2 + \frac{\mu}{4}\|y^*(x_t) - y_t\|^2 + \frac{\mu}{4}\|y_t - \tilde{y}_{t+1}\|^2. \tag{53}$$

By plugging the inequalities (52) and (53) into (50), we obtain

$$\frac{1}{2\eta_t\lambda}\|y_{t+1} - y^*(x_t)\|^2$$
$$\leq (\frac{1}{2\eta_t\lambda} - \frac{\mu}{4})\|y_t - y^*(x_t)\|^2 + (\frac{\eta_t}{2\lambda} + \frac{\mu}{4} + \frac{L_f}{2} - \frac{1}{\lambda})\|\tilde{y}_{t+1} - y_t\|^2 + \frac{2}{\mu}\|\nabla_y f(x_t, y_t) - w_t\|^2$$
$$\leq (\frac{1}{2\eta_t\lambda} - \frac{\mu}{4})\|y_t - y^*(x_t)\|^2 + (\frac{3L_f}{4} - \frac{1}{2\lambda})\|\tilde{y}_{t+1} - y_t\|^2 + \frac{2}{\mu}\|\nabla_y f(x_t, y_t) - w_t\|^2$$
$$= (\frac{1}{2\eta_t\lambda} - \frac{\mu}{4})\|y_t - y^*(x_t)\|^2 - (\frac{3}{8\lambda} + \frac{1}{8\lambda} - \frac{3L_f}{4})\|\tilde{y}_{t+1} - y_t\|^2 + \frac{2}{\mu}\|\nabla_y f(x_t, y_t) - w_t\|^2$$
$$\leq (\frac{1}{2\eta_t\lambda} - \frac{\mu}{4})\|y_t - y^*(x_t)\|^2 - \frac{3}{8\lambda}\|\tilde{y}_{t+1} - y_t\|^2 + \frac{2}{\mu}\|\nabla_y f(x_t, y_t) - w_t\|^2, \tag{54}$$

where the second inequality holds by $L_f \geq \mu$ and $0 < \eta_t \leq 1$, and the last inequality is due to $0 < \lambda \leq \frac{1}{6L_f}$. It implies that

$$\|y_{t+1} - y^*(x_t)\|^2 \leq (1 - \frac{\eta_t\mu\lambda}{2})\|y_t - y^*(x_t)\|^2 - \frac{3\eta_t}{4}\|\tilde{y}_{t+1} - y_t\|^2 + \frac{4\eta_t\lambda}{\mu}\|\nabla_y f(x_t, y_t) - w_t\|^2. \tag{55}$$

Next, we decompose the term $\|y_{t+1} - y^*(x_{t+1})\|^2$ as follows:

$$\|y_{t+1} - y^*(x_{t+1})\|^2 = \|y_{t+1} - y^*(x_t) + y^*(x_t) - y^*(x_{t+1})\|^2$$
$$= \|y_{t+1} - y^*(x_t)\|^2 + 2\langle y_{t+1} - y^*(x_t), y^*(x_t) - y^*(x_{t+1})\rangle + \|y^*(x_t) - y^*(x_{t+1})\|^2$$
$$\leq (1 + \frac{\eta_t\mu\lambda}{4})\|y_{t+1} - y^*(x_t)\|^2 + (1 + \frac{4}{\eta_t\mu\lambda})\|y^*(x_t) - y^*(x_{t+1})\|^2$$
$$\leq (1 + \frac{\eta_t\mu\lambda}{4})\|y_{t+1} - y^*(x_t)\|^2 + (1 + \frac{4}{\eta_t\mu\lambda})\kappa^2\|x_t - x_{t+1}\|^2, \tag{56}$$

where the first inequality holds by Cauchy-Schwarz inequality and Young's inequality, and the last inequality is due to Lemma 1.

By combining the above inequalities (55) and (56), we have

$$\|y_{t+1} - y^*(x_{t+1})\|^2 \leq (1 + \frac{\eta_t\mu\lambda}{4})(1 - \frac{\eta_t\mu\lambda}{2})\|y_t - y^*(x_t)\|^2 - (1 + \frac{\eta_t\mu\lambda}{4})\frac{3\eta_t}{4}\|\tilde{y}_{t+1} - y_t\|^2$$
$$+ (1 + \frac{\eta_t\mu\lambda}{4})\frac{4\eta_t\lambda}{\mu}\|\nabla_y f(x_t, y_t) - w_t\|^2 + (1 + \frac{4}{\eta_t\mu\lambda})\kappa^2\|x_t - x_{t+1}\|^2. \tag{57}$$

Since $0 < \eta_t \le 1$, $0 < \lambda \le \frac{1}{6L_f}$ and $L_f \ge \mu$, we have $\lambda \le \frac{1}{6L_f} \le \frac{1}{6\mu}$ and $\eta_t \le 1 \le \frac{1}{6\mu\lambda}$. Then we obtain

$$(1 + \frac{\eta_t\mu\lambda}{4})(1 - \frac{\eta_t\mu\lambda}{2}) = 1 - \frac{\eta_t\mu\lambda}{2} + \frac{\eta_t\mu\lambda}{4} - \frac{\eta_t^2\mu^2\lambda^2}{8} \le 1 - \frac{\eta_t\mu\lambda}{4},$$

$$-(1 + \frac{\eta_t\mu\lambda}{4})\frac{3\eta_t}{4} \le -\frac{3\eta_t}{4},$$

$$(1 + \frac{\eta_t\mu\lambda}{4})\frac{4\eta_t\lambda}{\mu} \le (1 + \frac{1}{24})\frac{4\eta_t\lambda}{\mu} = \frac{25\eta_t\lambda}{6\mu},$$

$$(1 + \frac{4}{\eta_t\mu\lambda})\kappa^2 = \kappa^2 + \frac{4\kappa^2}{\eta_t\mu\lambda} \le \frac{\kappa^2}{6\eta_t\mu\lambda} + \frac{4\kappa^2}{\eta_t\mu\lambda} = \frac{25\kappa^2}{6\eta_t\mu\lambda}. \tag{58}$$

Thus we have

$$\|y_{t+1} - y^*(x_{t+1})\|^2 \le (1 - \frac{\eta_t\mu\lambda}{4})\|y_t - y^*(x_t)\|^2 - \frac{3\eta_t}{4}\|\tilde{y}_{t+1} - y_t\|^2$$

$$+ \frac{25\eta_t\lambda}{6\mu}\|\nabla_y f(x_t, y_t) - w_t\|^2 + \frac{25\kappa^2}{6\eta_t\mu\lambda}\|x_t - x_{t+1}\|^2. \tag{59}$$

$\square$

**Theorem 4.** *(Restatement of Theorem 1) Suppose the sequence $\{x_t, y_t\}_{t=1}^T$ be generated from Algorithm 1 using stochastic partial gradients (i.e., SMDA algorithm). Let $0 < \eta = \eta_t \le 1$, $0 < \gamma = \gamma_t \le \min(\frac{3\rho}{4L}, \frac{9\eta\rho\mu\lambda}{800\kappa^2}, \frac{2\eta\mu\rho\lambda}{25L_f^2})$ and $0 < \lambda \le \frac{1}{6L_f}$, we have*

$$\frac{1}{T}\sum_{t=1}^T \mathbb{E}\|\mathcal{G}_t\| \le \frac{4\sqrt{2(\tilde{F}(x_1) - F^*)}}{\sqrt{3T\gamma\rho}} + \frac{4\sqrt{2}\Delta_1}{\sqrt{3T\gamma\rho}} + \frac{10\sigma}{\sqrt{3b\rho}} + \frac{20\sigma\sqrt{\eta\lambda}}{3\sqrt{\gamma\rho\mu b}}, \tag{60}$$

*where $L = L_f(1 + \kappa)$, $\tilde{F}(x) = \Phi(x) + g(x)$ and $\Delta_1 = \|y_1 - y^*(x_1)\|$.*

*Proof.* According to the above Lemma 1, the function $\Phi(x)$ has $L$-Lipschitz continuous gradient. Let $\tilde{\mathcal{G}}_t = \frac{1}{\gamma_t}(x_t - x_{t+1})$, we have

$$\Phi(x_{t+1}) \le \Phi(x_t) + \langle \nabla\Phi(x_t), x_{t+1} - x_t \rangle + \frac{L}{2}\|x_{t+1} - x_t\|^2$$

$$= \Phi(x_t) - \gamma_t\langle \nabla\Phi(x_t), \tilde{\mathcal{G}}_t \rangle + \frac{\gamma_t^2 L}{2}\|\tilde{\mathcal{G}}_t\|^2$$

$$= \Phi(x_t) - \gamma_t\langle v_t, \tilde{\mathcal{G}}_t \rangle + \gamma_t\langle v_t - \nabla\Phi(x_t), \tilde{\mathcal{G}}_t \rangle + \frac{\gamma_t^2 L}{2}\|\tilde{\mathcal{G}}_t\|^2$$

$$\le \Phi(x_t) - \gamma_t\rho\|\tilde{\mathcal{G}}_t\|^2 - g(x_{t+1}) + g(x_t) + \gamma_t\langle v_t - \nabla\Phi(x_t), \tilde{\mathcal{G}}_t \rangle + \frac{\gamma_t^2 L}{2}\|\tilde{\mathcal{G}}_t\|^2$$

$$\le \Phi(x_t) + (\frac{\gamma_t^2 L}{2} - \frac{3\gamma_t\rho}{4})\|\tilde{\mathcal{G}}_t\|^2 - g(x_{t+1}) + g(x_t) + \frac{\gamma_t}{\rho}\|v_t - \nabla\Phi(x_t)\|^2, \tag{61}$$

where the second last inequality holds by the above Lemma 4, and the last inequality holds by the following inequality

$$\langle v_t - \nabla\Phi(x_t), \tilde{\mathcal{G}}_t \rangle \le \|v_t - \nabla\Phi(x_t)\|\|\tilde{\mathcal{G}}_t\|$$

$$\le \frac{1}{\rho}\|v_t - \nabla\Phi(x_t)\|^2 + \frac{\rho}{4}\|\tilde{\mathcal{G}}_t\|^2. \tag{62}$$

According to the above Lemma 1 and Assumption 1, we have

$$\|v_t - \nabla\Phi(x_t)\|^2 = \|v_t - \nabla f(x_t, y^*(x_t))\|^2$$

$$= \|v_t - \nabla_x f(x_t, y_t) + \nabla_x f(x_t, y_t) - \nabla_x f(x_t, y^*(x_t))\|^2$$

$$\le 2\|v_t - \nabla_x f(x_t, y_t)\|^2 + 2\|\nabla_x f(x_t, y_t) - \nabla_x f(x_t, y^*(x_t))\|^2$$

$$\le 2\|v_t - \nabla_x f(x_t, y_t)\|^2 + 2L_f^2\|y_t - y^*(x_t)\|^2. \tag{63}$$

Let $\tilde{F}(x) = \Phi(x) + g(x)$, plugging (63) into (61), we have

$$\tilde{F}(x_{t+1}) \le \tilde{F}(x_t) + (\frac{\gamma_t^2 L}{2} - \frac{3\gamma_t\rho}{4})\|\tilde{\mathcal{G}}_t\|^2 + \frac{2\gamma_t}{\rho}\|v_t - \nabla_x f(x_t, y_t)\|^2 + \frac{2L_f^2\gamma_t}{\rho}\|y_t - y^*(x_t)\|^2$$

$$\le \tilde{F}(x_t) - \frac{3\gamma_t\rho}{8}\|\tilde{\mathcal{G}}_t\|^2 + \frac{2\gamma_t}{\rho}\|v_t - \nabla_x f(x_t, y_t)\|^2 + \frac{2L_f^2\gamma_t}{\rho}\|y_t - y^*(x_t)\|^2, \tag{64}$$

where the last inequality is due to $0 < \gamma_t \le \frac{3\rho}{4L}$. According to Lemma 5, the difference between $\tilde{\mathcal{G}}_t$ and $\mathcal{G}_t$ are bounded, we have

$$
\begin{aligned}
\|\mathcal{G}_t\|^2 &\le 2\|\tilde{\mathcal{G}}_t\|^2 + 2\|\tilde{\mathcal{G}}_t - \mathcal{G}_t\|^2 \\
&\le 2\|\tilde{\mathcal{G}}_t\|^2 + \frac{2}{\rho^2}\|v_t - \nabla\Phi(x_t)\|^2 \\
&\le 2\|\tilde{\mathcal{G}}_t\|^2 + \frac{4}{\rho^2}\|v_t - \nabla_x f(x_t, y_t)\|^2 + \frac{4L_f^2}{\rho^2}\|y_t - y^*(x_t)\|^2.
\end{aligned}
\tag{65}
$$

Thus we have

$$
-\|\tilde{\mathcal{G}}_t\|^2 \le -\frac{1}{2}\|\mathcal{G}_t\|^2 + \frac{2}{\rho^2}\|v_t - \nabla_x f(x_t, y_t)\|^2 + \frac{2L_f^2}{\rho^2}\|y_t - y^*(x_t)\|^2.
\tag{66}
$$

By plugging (66) into (61), we have

$$
\begin{aligned}
\tilde{F}(x_{t+1}) &\le \tilde{F}(x_t) - \frac{3\gamma_t\rho}{16}\|\mathcal{G}_t\|^2 + \frac{3\gamma_t\rho}{8}\Big(\frac{2}{\rho^2}\|v_t - \nabla_x f(x_t, y_t)\|^2 + \frac{2L_f^2}{\rho^2}\|y_t - y^*(x_t)\|^2\Big) \\
&\quad + \frac{2\gamma_t}{\rho}\|v_t - \nabla_x f(x_t, y_t)\|^2 + \frac{2L_f^2\gamma_t}{\rho}\|y_t - y^*(x_t)\|^2 \\
&= \tilde{F}(x_t) - \frac{3\gamma_t\rho}{16}\|\mathcal{G}_t\|^2 + \frac{11\gamma_t}{4\rho}\|v_t - \nabla_x f(x_t, y_t)\|^2 + \frac{11L_f^2\gamma_t}{4\rho}\|y_t - y^*(x_t)\|^2.
\end{aligned}
\tag{67}
$$

Next, we define a useful Lyapunov function, for any $t \ge 1$

$$
\Omega_t = \tilde{F}(x_t) + \|y_t - y^*(x_t)\|^2.
\tag{68}
$$

According to Lemma 6, we have

$$
\begin{aligned}
\|y_{t+1} - y^*(x_{t+1})\|^2 - \|y_t - y^*(x_t)\|^2 &\le -\frac{\eta_t\mu\lambda}{4}\|y_t - y^*(x_t)\|^2 - \frac{3\eta_t}{4}\|\tilde{y}_{t+1} - y_t\|^2 \\
&\quad + \frac{25\eta_t\lambda}{6\mu}\|\nabla_y f(x_t, y_t) - w_t\|^2 + \frac{25\kappa^2}{6\eta_t\mu\lambda}\|x_t - x_{t+1}\|^2.
\end{aligned}
\tag{69}
$$

Similarly, we have

$$
\begin{aligned}
\|\tilde{\mathcal{G}}_t\|^2 &\le 2\|\mathcal{G}_t\|^2 + 2\|\tilde{\mathcal{G}}_t - \mathcal{G}_t\|^2 \\
&\le 2\|\mathcal{G}_t\|^2 + \frac{2}{\rho^2}\|v_t - \nabla\Phi(x_t)\|^2 \\
&\le 2\|\mathcal{G}_t\|^2 + \frac{4}{\rho^2}\|v_t - \nabla_x f(x_t, y_t)\|^2 + \frac{4L_f^2}{\rho^2}\|y_t - y^*(x_t)\|^2.
\end{aligned}
\tag{70}
$$

Then we have

$$
\begin{aligned}
\Omega_{t+1} - \Omega_t &= \tilde{F}(x_{t+1}) - \tilde{F}(x_t) + \|y_{t+1} - y^*(x_{t+1})\|^2 - \|y_t - y^*(x_t)\|^2 \\
&\le -\frac{3\gamma_t\rho}{16}\|\mathcal{G}_t\|^2 + \frac{11\gamma_t}{4\rho}\|v_t - \nabla_x f(x_t, y_t)\|^2 + \frac{11L_f^2\gamma_t}{4\rho}\|y_t - y^*(x_t)\|^2 - \frac{\eta_t\mu\lambda}{4}\|y_t - y^*(x_t)\|^2 \\
&\quad - \frac{3\eta_t}{4}\|\tilde{y}_{t+1} - y_t\|^2 + \frac{25\eta_t\lambda}{6\mu}\|\nabla_y f(x_t, y_t) - w_t\|^2 + \frac{25\kappa^2}{6\eta_t\mu\lambda}\|x_t - x_{t+1}\|^2 \\
&= -\frac{3\gamma_t\rho}{16}\|\mathcal{G}_t\|^2 + \frac{11\gamma_t}{4\rho}\|v_t - \nabla_x f(x_t, y_t)\|^2 + \frac{11L_f^2\gamma_t}{4\rho}\|y_t - y^*(x_t)\|^2 - \frac{\eta_t\mu\lambda}{4}\|y_t - y^*(x_t)\|^2 \\
&\quad - \frac{3\eta_t}{4}\|\tilde{y}_{t+1} - y_t\|^2 + \frac{25\eta_t\lambda}{6\mu}\|\nabla_y f(x_t, y_t) - w_t\|^2 + \frac{25\kappa^2\gamma_t^2}{6\eta_t\mu\lambda}\|\tilde{\mathcal{G}}_t\|^2 \\
&\le -\Big(\frac{3\gamma_t\rho}{16} - \frac{25\kappa^2\gamma_t^2}{3\eta_t\mu\lambda}\Big)\|\mathcal{G}_t\|^2 + \Big(\frac{11\gamma_t}{4\rho} + \frac{50\kappa^2\gamma_t^2}{3\eta_t\mu\lambda\rho^2}\Big)\|v_t - \nabla_x f(x_t, y_t)\|^2 \\
&\quad + \Big(\frac{11L_f^2\gamma_t}{4\rho} + \frac{50\kappa^2\gamma_t^2 L_f^2}{3\eta_t\mu\lambda\rho^2} - \frac{\eta_t\mu\lambda}{4}\Big)\|y_t - y^*(x_t)\|^2 - \frac{3\eta_t}{4}\|\tilde{y}_{t+1} - y_t\|^2 \\
&\quad + \frac{25\eta_t\lambda}{6\mu}\|\nabla_y f(x_t, y_t) - w_t\|^2,
\end{aligned}
\tag{71}
$$

where the last inequality holds by the inequality (70).

Let $\gamma = \gamma_t$ and $\eta = \eta_t$ for all $t \geq 1$. By using $0 < \gamma \leq \frac{9\eta\rho\mu\lambda}{800\kappa^2}$, we have

$$\frac{3\gamma\rho}{32} \geq \frac{50\kappa^2\gamma^2}{6\eta\lambda\mu}, \quad \frac{3L_f^2\gamma}{8\rho} \geq \frac{50\kappa^2\gamma^2 L_f^2}{3\eta_t\mu\lambda\rho^2}, \quad \frac{3\gamma}{8\rho} \geq \frac{50\kappa^2\gamma^2}{3\eta_t\mu\lambda\rho^2}. \tag{72}$$

Let $\frac{\eta\mu\lambda}{4} \geq \frac{25L_f^2\gamma}{8\rho}$, we have $0 < \gamma \leq \frac{2\eta\mu\rho\lambda}{25L_f^2}$. Given $0 < \gamma \leq \min(\frac{9\eta\rho\mu\lambda}{800\kappa^2}, \frac{2\eta\mu\rho\lambda}{25L_f^2})$, we have

$$\Omega_{t+1} - \Omega_t \leq -\frac{3\gamma\rho}{32}\|\mathcal{G}_t\|^2 + \frac{25\gamma}{8\rho}\|v_t - \nabla_x f(x_t, y_t)\|^2 + \frac{25\eta\lambda}{6\mu}\|w_t - \nabla_y f(x_t, y_t)\|^2. \tag{73}$$

Thus we have

$$\mathbb{E}\|\mathcal{G}_t\|^2 \leq \frac{32\mathbb{E}(\Omega_t - \Omega_{t+1})}{3\gamma\rho} + \frac{100}{3\rho^2}\mathbb{E}\|v_t - \nabla_x f(x_t, y_t)\|^2 + \frac{400\eta\lambda}{9\gamma\rho\mu}\mathbb{E}\|\nabla_y f(x_t, y_t) - w_t\|^2$$

$$\leq \frac{32\mathbb{E}(\Omega_t - \Omega_{t+1})}{3\gamma\rho} + \frac{100\sigma^2}{3\rho^2 b} + \frac{400\eta\lambda\sigma^2}{9\gamma\rho\mu b}, \tag{74}$$

where the last inequality holds by Assumption 2 and $v_t = \nabla_x f_{\mathcal{B}_t}(x_t, y_t) = \frac{1}{b}\sum_{i \in \mathcal{B}_t} \nabla_x f(x_t, y_t, \xi_t^i)$, $w_t = \nabla_y f_{\mathcal{B}_t}(x_t, y_t) = \frac{1}{b}\sum_{i \in \mathcal{B}_t} \nabla_y f(x_t, y_t, \xi_t^i)$.

Taking average over $t = 1, 2, \cdots, T$ on both sides of the above inequality (74), we have

$$\frac{1}{T}\sum_{t=1}^{T}\mathbb{E}\|\mathcal{G}_t\|^2 \leq \frac{32\mathbb{E}(\Omega_1 - \Omega_{T+1})}{3T\gamma\rho} + \frac{100\sigma^2}{3\rho^2 b} + \frac{400\eta\lambda\sigma^2}{9\gamma\rho\mu b}$$

$$= \frac{32(\tilde{F}(x_1) + \|y_1 - y^*(x_1)\|^2)}{3T\gamma\rho} - \frac{32\mathbb{E}(\tilde{F}(x_{T+1}) + \|y_{T+1} - y^*(x_{T+1})\|^2)}{3T\gamma\rho}$$

$$+ \frac{100\sigma^2}{3\rho^2 b} + \frac{400\eta\lambda\sigma^2}{9\gamma\rho\mu b}$$

$$\leq \frac{32(\tilde{F}(x_1) - F^*)}{3T\gamma\rho} + \frac{32\Delta_1^2}{3T\gamma\rho} + \frac{100\sigma^2}{3\rho^2 b} + \frac{400\eta\lambda\sigma^2}{9\gamma\rho\mu b}, \tag{75}$$

where the last inequality holds by Assumption 5 and $\Delta_1 = \|y_1 - y^*(x_1)\|$. By using Jensen's inequality, we have

$$\frac{1}{T}\sum_{t=1}^{T}\mathbb{E}\|\mathcal{G}_t\| \leq \left(\frac{1}{T}\sum_{t=1}^{T}\mathbb{E}\|\mathcal{G}_t\|^2\right)^{\frac{1}{2}}$$

$$\leq \frac{4\sqrt{2(\tilde{F}(x_1) - F^*)}}{\sqrt{3T\gamma\rho}} + \frac{4\sqrt{2}\Delta_1}{\sqrt{3T\gamma\rho}} + \frac{10\sigma}{\sqrt{3b\rho}} + \frac{20\sigma\sqrt{\eta\lambda}}{3\sqrt{\gamma\rho\mu b}}, \tag{76}$$

where the last inequality is due to the inequality $(\sum_{i=1}^{4} a_i)^{\frac{1}{2}} \leq \sum_{i=1}^{4} a_i^{\frac{1}{2}}$ for all $a_i \geq 0$, $i = 1, 2, 3, 4$.

$\square$

**Theorem 5.** *(Restatement of Theorem 2) Suppose the sequence $\{x_t, y_t\}_{t=1}^{T}$ be generated from Algorithm 1 using deterministic partial gradients (i.e., MDA algorithm). Let $0 < \eta = \eta_t \leq 1$, $0 < \gamma = \gamma_t \leq \min(\frac{3\rho}{4L}, \frac{9\eta\rho\mu\lambda}{800\kappa^2}, \frac{2\eta\mu\rho\lambda}{25L_f^2})$ and $0 < \lambda \leq \frac{1}{6L_f}$, we have*

$$\frac{1}{T}\sum_{t=1}^{T}\|\mathcal{G}_t\| \leq \frac{4\sqrt{2(\tilde{F}(x_1) - F^*)}}{\sqrt{3T\gamma\rho}} + \frac{4\sqrt{2}\Delta_1}{\sqrt{3T\gamma\rho}}, \tag{77}$$

*where $L = L_f(1 + \kappa)$, $\tilde{F}(x) = \Phi(x) + g(x)$ and $\Delta_1 = \|y_1 - y^*(x_1)\|$.*

*Proof.* This proof can follow the proof of Theorem 1. Since the MDA algorithm uses the deterministic partial gradients $v_t = \nabla_x f(x_t, y_t)$ and $w_t = \nabla_y f(x_t, y_t)$, we have $\sigma = 0$. $\square$

## A.2 Convergence Analysis of the VR-SMDA Algorithm

In the subsection, we provide the convergence analysis of the VR-SMDA algorithm.

**Lemma 7.** *Suppose the stochastic gradients $v_t$ and $w_t$ be generated from Algorithm 2, we have*

$$\mathbb{E}\|\nabla_x f(x_t, y_t) - v_t\|^2 \leq \frac{L_f^2}{b_1}\sum_{i=(n_t-1)q}^{t-1}\left(\mathbb{E}\|x_{i+1} - x_i\|^2 + \mathbb{E}\|y_{i+1} - y_i\|^2\right) + \frac{\sigma^2}{b}, \tag{78}$$

$$\mathbb{E}\|\nabla_y f(x_t, y_t) - w_t\|^2 \leq \frac{L_f^2}{b_1} \sum_{i=(n_t-1)q}^{t-1} \left(\mathbb{E}\|x_{i+1} - x_i\|^2 + \mathbb{E}\|y_{i+1} - y_i\|^2\right) + \frac{\sigma^2}{b}. \tag{79}$$

*Proof.* We first prove the inequality (78). According to the definition of $v_{t-1}$ in Algorithm 2, we have

$$v_t - v_{t-1} = \nabla_x f_{\mathcal{I}_t}(x_t, y_t) - \nabla_x f_{\mathcal{I}_t}(x_{t-1}, y_{t-1}). \tag{80}$$

Then we have

$$\mathbb{E}\|\nabla_x f(x_t, y_t) - v_t\|^2$$
$$= \mathbb{E}\|\nabla_x f(x_t, y_t) - v_{t-1} - (v_t - v_{t-1})\|^2$$
$$= \mathbb{E}\|\nabla_x f(x_t, y_t) - v_{t-1} - \nabla_x f_{\mathcal{I}_t}(x_t, y_t) + \nabla_x f_{\mathcal{I}_t}(x_{t-1}, y_{t-1})\|^2$$
$$= \mathbb{E}\|\nabla_x f(x_{t-1}, y_{t-1}) - v_{t-1} + \nabla_x f(x_t, y_t) - \nabla_x f(x_{t-1}, y_{t-1}) - \nabla_x f_{\mathcal{I}_t}(x_t, y_t) + \nabla_x f_{\mathcal{I}_t}(x_{t-1}, y_{t-1})\|^2$$
$$= \mathbb{E}\|\nabla_x f(x_{t-1}, y_{t-1}) - v_{t-1}\|^2 + \mathbb{E}\|\nabla_x f(x_t, y_t) - \nabla_x f(x_{t-1}, y_{t-1})$$
$$\quad - \left(\nabla_x f_{\mathcal{I}_t}(x_t, y_t) - \nabla_x f_{\mathcal{I}_t}(x_{t-1}, y_{t-1})\right)\|^2$$
$$= \mathbb{E}\|\nabla_x f(x_{t-1}, y_{t-1}) - v_{t-1}\|^2 + \frac{1}{b_1}\mathbb{E}\|\nabla_x f(x_t, y_t) - \nabla_x f(x_{t-1}, y_{t-1})$$
$$\quad - \left(\nabla_x f(x_t, y_t; \xi_t^1) - \nabla_x f(x_{t-1}, y_{t-1}; \xi_t^1)\right)\|^2$$
$$\leq \mathbb{E}\|\nabla_x f(x_{t-1}, y_{t-1}) - v_{t-1}\|^2 + \frac{1}{b_1}\mathbb{E}\|\nabla_x f(x_t, y_t; \xi_t^1) - \nabla_x f(x_{t-1}, y_{t-1}; \xi_t^1)\|^2$$
$$\leq \mathbb{E}\|\nabla_x f(x_{t-1}, y_{t-1}) - v_{t-1}\|^2 + \frac{L_f^2}{b_1}\left(\|x_t - x_{t-1}\|^2 + \|y_t - y_{t-1}\|^2\right), \tag{81}$$

where the fourth equality follows by $\mathbb{E}_{\mathcal{I}_t}\left[\nabla_x f(x_t, y_t) - \nabla_x f(x_{t-1}, y_{t-1}) - \left(\nabla_x f_{\mathcal{I}_t}(x_t, y_t) - \nabla_x f_{\mathcal{I}_t}(x_{t-1}, y_{t-1})\right)\right] = 0$; the fifth equality holds by Lemma 3 and $\nabla_x f_{\mathcal{I}_t}(x_t, y_t) = \frac{1}{b_1}\sum_{i \in \mathcal{I}_t} \nabla_x f(x_t, y_t, \xi_t^i)$, $\nabla_x f_{\mathcal{I}_t}(x_{t-1}, y_{t-1}) = \frac{1}{b_1}\sum_{i \in \mathcal{I}_t} \nabla_x f(x_{t-1}, y_{t-1}, \xi_t^i)$; the second last inequality holds by the inequality $\mathbb{E}\|\zeta - \mathbb{E}[\zeta]\|^2 \leq \mathbb{E}\|\zeta\|^2$; the last inequality is due to Assumption 1.

Throughout the paper, let $n_t = [t/q]$ such that $(n_t - 1)q \leq t \leq n_t q - 1$. Telescoping (81) over $t$ from $(n_t - 1)q + 1$ to $t$, we have

$$\mathbb{E}\|\nabla_x f(x_t, y_t) - v_t\|^2 \leq \frac{L_f^2}{b_1} \sum_{i=(n_t-1)q}^{t-1} \left(\mathbb{E}\|x_{i+1} - x_i\|^2 + \mathbb{E}\|y_{i+1} - y_i\|^2\right)$$
$$\quad + \mathbb{E}\|\nabla_x f(x_{(n_t-1)q}, y_{(n_t-1)q}) - v_{(n_t-1)q}\|^2$$
$$\leq \frac{L_f^2}{b_1} \sum_{i=(n_t-1)q}^{t-1} \left(\mathbb{E}\|x_{i+1} - x_i\|^2 + \mathbb{E}\|y_{i+1} - y_i\|^2\right) + \frac{\sigma^2}{b}, \tag{82}$$

where the last inequality is due to Assumption 2 and $v_{(n_t-1)q} = \frac{1}{b}\sum_{i \in \mathcal{B}_{(n_t-1)q}} \nabla_x f(x_{(n_t-1)q}, y_{(n_t-1)q}, \xi_{(n_t-1)q}^i)$. Similarly, we can obtain

$$\mathbb{E}\|\nabla_y f(x_t, y_t) - w_t\|^2 \leq \frac{L_f^2}{b_1} \sum_{i=(n_t-1)q}^{t-1} \left(\mathbb{E}\|x_{i+1} - x_i\|^2 + \mathbb{E}\|y_{i+1} - y_i\|^2\right) + \frac{\sigma^2}{b}. \tag{83}$$

□

**Theorem 6.** *(Restatement of Theorem 3) Suppose the sequence $\{x_t, y_t\}_{t=1}^T$ be generated from Algorithm 2. Let $b_1 = q$, $0 < \eta = \eta_t \leq 1$, $0 < \gamma = \gamma_t \leq \min(\frac{3\rho}{4L}, \frac{\eta\mu\lambda\rho}{38L_f^2}, \frac{3\rho}{19L_f^2\eta}, \frac{\rho\eta}{8}, \frac{9\rho\eta\mu\lambda}{400\kappa^2})$ and $0 < \lambda \leq \min(\frac{1}{6L_f}, \frac{9\mu}{100\eta^2 L_f^2})$, we have*

$$\frac{1}{T}\sum_{t=1}^T \mathbb{E}\|\mathcal{G}_t\| \leq \frac{4\sqrt{2(\tilde{F}(x_1) - F^*)}}{\sqrt{3T\gamma\rho}} + \frac{4\sqrt{2}\Delta_1}{\sqrt{3T\gamma\rho}} + \frac{2\sqrt{2}\sigma}{\sqrt{\gamma\rho\eta b L_f}}, \tag{84}$$

*where $L = L_f(1 + \kappa)$, $\tilde{F}(x) = \Phi(x) + g(x)$ and $\Delta_1 = \|y_1 - y^*(x_1)\|$.*

*Proof.* This proof is similar to the proof of Theorem 1. According to the above Lemma 1, the function $\Phi(x)$ has $L$-Lipschitz continuous gradient. Let $\tilde{\mathcal{G}}_t = \frac{1}{\gamma_t}(x_t - x_{t+1})$, we have

$$\Phi(x_{t+1}) \leq \Phi(x_t) + \langle \nabla\Phi(x_t), x_{t+1} - x_t \rangle + \frac{L}{2}\|x_{t+1} - x_t\|^2$$

$$= \Phi(x_t) - \gamma_t \langle \nabla\Phi(x_t), \tilde{\mathcal{G}}_t \rangle + \frac{\gamma_t^2 L}{2}\|\tilde{\mathcal{G}}_t\|^2$$

$$= \Phi(x_t) - \gamma_t \langle v_t, \tilde{\mathcal{G}}_t \rangle + \gamma_t \langle v_t - \nabla\Phi(x_t), \tilde{\mathcal{G}}_t \rangle + \frac{\gamma_t^2 L}{2}\|\tilde{\mathcal{G}}_t\|^2$$

$$\leq \Phi(x_t) - \gamma_t \rho\|\tilde{\mathcal{G}}_t\|^2 - g(x_{t+1}) + g(x_t) + \gamma_t \langle v_t - \nabla\Phi(x_t), \tilde{\mathcal{G}}_t \rangle + \frac{\gamma_t^2 L}{2}\|\tilde{\mathcal{G}}_t\|^2$$

$$\leq \Phi(x_t) + (\frac{\gamma_t^2 L}{2} - \frac{3\gamma_t \rho}{4})\|\tilde{\mathcal{G}}_t\|^2 - g(x_{t+1}) + g(x_t) + \frac{\gamma_t}{\rho}\|v_t - \nabla\Phi(x_t)\|^2, \tag{85}$$

where the second last inequality holds by the above Lemma 4, and the last inequality holds by the following inequality

$$\langle v_t - \nabla\Phi(x_t), \tilde{\mathcal{G}}_t \rangle \leq \|v_t - \nabla\Phi(x_t)\|\|\tilde{\mathcal{G}}_t\|$$

$$\leq \frac{1}{\rho}\|v_t - \nabla\Phi(x_t)\|^2 + \frac{\rho}{4}\|\tilde{\mathcal{G}}_t\|^2. \tag{86}$$

According to the above Lemma 1 and Assumption 1, we have

$$\|v_t - \nabla\Phi(x_t)\|^2 = \|v_t - \nabla f(x_t, y^*(x_t))\|^2$$

$$= \|v_t - \nabla_x f(x_t, y_t) + \nabla_x f(x_t, y_t) - \nabla_x f(x_t, y^*(x_t))\|^2$$

$$\leq 2\|v_t - \nabla_x f(x_t, y_t)\|^2 + 2\|\nabla_x f(x_t, y_t) - \nabla_x f(x_t, y^*(x_t))\|^2$$

$$\leq 2\|v_t - \nabla_x f(x_t, y_t)\|^2 + 2L_f^2\|y_t - y^*(x_t)\|^2. \tag{87}$$

Let $\tilde{F}(x) = \Phi(x) + g(x)$, plugging (87) into (85), we have

$$\tilde{F}(x_{t+1}) \leq \tilde{F}(x_t) + (\frac{\gamma_t^2 L}{2} - \frac{3\gamma_t \rho}{4})\|\tilde{\mathcal{G}}_t\|^2 + \frac{2\gamma_t}{\rho}\|v_t - \nabla_x f(x_t, y_t)\|^2 + \frac{2L_f^2 \gamma_t}{\rho}\|y_t - y^*(x_t)\|^2$$

$$\leq \tilde{F}(x_t) - \frac{3\gamma_t \rho}{8}\|\tilde{\mathcal{G}}_t\|^2 + \frac{2\gamma_t}{\rho}\|v_t - \nabla_x f(x_t, y_t)\|^2 + \frac{2L_f^2 \gamma_t}{\rho}\|y_t - y^*(x_t)\|^2$$

$$= \tilde{F}(x_t) - \frac{3\gamma_t \rho}{16}\|\tilde{\mathcal{G}}_t\|^2 - \frac{3\rho}{16\gamma_t}\|x_{t+1} - x_t\|^2 + \frac{2\gamma_t}{\rho}\|v_t - \nabla_x f(x_t, y_t)\|^2 + \frac{2L_f^2 \gamma_t}{\rho}\|y_t - y^*(x_t)\|^2, \tag{88}$$

where the second inequality is due to $0 < \gamma_t \leq \frac{3\rho}{4L}$ and the last equality holds by $\tilde{\mathcal{G}}_t = \frac{1}{\gamma_t}(x_{t+1} - x_t)$. By using Lemma 5, the difference between $\tilde{\mathcal{G}}_t$ and $\mathcal{G}_t$ are bounded, we have

$$\|\mathcal{G}_t\|^2 \leq 2\|\tilde{\mathcal{G}}_t\|^2 + 2\|\tilde{\mathcal{G}}_t - \mathcal{G}_t\|^2$$

$$\leq 2\|\tilde{\mathcal{G}}_t\|^2 + \frac{2}{\rho^2}\|v_t - \nabla\Phi(x_t)\|^2$$

$$\leq 2\|\tilde{\mathcal{G}}_t\|^2 + \frac{4}{\rho^2}\|v_t - \nabla_x f(x_t, y_t)\|^2 + \frac{4L_f^2}{\rho^2}\|y_t - y^*(x_t)\|^2. \tag{89}$$

Thus we have

$$-\|\tilde{\mathcal{G}}_t\|^2 \leq -\frac{1}{2}\|\mathcal{G}_t\|^2 + \frac{2}{\rho^2}\|v_t - \nabla_x f(x_t, y_t)\|^2 + \frac{2L_f^2}{\rho^2}\|y_t - y^*(x_t)\|^2. \tag{90}$$

By plugging (90) into (85), we have

$$\tilde{F}(x_{t+1}) \leq \tilde{F}(x_t) - \frac{3\gamma_t \rho}{32}\|\mathcal{G}_t\|^2 + \frac{3\gamma_t \rho}{16}\left(\frac{2}{\rho^2}\|v_t - \nabla_x f(x_t, y_t)\|^2 + \frac{2L_f^2}{\rho^2}\|y_t - y^*(x_t)\|^2\right)$$

$$- \frac{3\rho}{16\gamma_t}\|x_{t+1} - x_t\|^2 + \frac{2\gamma_t}{\rho}\|v_t - \nabla_x f(x_t, y_t)\|^2 + \frac{2L_f^2 \gamma_t}{\rho}\|y_t - y^*(x_t)\|^2$$

$$= \tilde{F}(x_t) - \frac{3\gamma_t \rho}{32}\|\mathcal{G}_t\|^2 - \frac{3\rho}{16\gamma_t}\|x_{t+1} - x_t\|^2 + \frac{19\gamma_t}{8\rho}\|v_t - \nabla_x f(x_t, y_t)\|^2 + \frac{19L_f^2 \gamma_t}{8\rho}\|y_t - y^*(x_t)\|^2. \tag{91}$$

Next, we define a useful Lyapunov function, for any $t \geq 1$

$$\Omega_t = \tilde{F}(x_t) + \|y_t - y^*(x_t)\|^2. \tag{92}$$

According to Lemma 6, we have

$$\|y_{t+1} - y^*(x_{t+1})\|^2 - \|y_t - y^*(x_t)\|^2 \leq -\frac{\eta_t \mu \lambda}{4}\|y_t - y^*(x_t)\|^2 - \frac{3\eta_t}{4}\|\tilde{y}_{t+1} - y_t\|^2$$
$$+ \frac{25\eta_t \lambda}{6\mu}\|\nabla_y f(x_t, y_t) - w_t\|^2 + \frac{25\kappa^2}{6\eta_t \mu \lambda}\|x_t - x_{t+1}\|^2. \tag{93}$$

Then we have

$$\Omega_{t+1} - \Omega_t = \tilde{F}(x_{t+1}) - \tilde{F}(x_t) + \|y_{t+1} - y^*(x_{t+1})\|^2 - \|y_t - y^*(x_t)\|^2$$

$$\leq -\frac{3\gamma_t \rho}{32}\|\mathcal{G}_t\|^2 - \frac{3\rho}{16\gamma_t}\|x_{t+1} - x_t\|^2 + \frac{19\gamma_t}{8\rho}\|v_t - \nabla_x f(x_t, y_t)\|^2 + \frac{19 L_f^2 \gamma_t}{8\rho}\|y_t - y^*(x_t)\|^2$$

$$- \frac{\eta_t \mu \lambda}{4}\|y_t - y^*(x_t)\|^2 - \frac{3\eta_t}{4}\|\tilde{y}_{t+1} - y_t\|^2 + \frac{25\eta_t \lambda}{6\mu}\|\nabla_y f(x_t, y_t) - w_t\|^2 + \frac{25\kappa^2}{6\eta_t \mu \lambda}\|x_t - x_{t+1}\|^2$$

$$= -\frac{3\gamma_t \rho}{32}\|\mathcal{G}_t\|^2 - \left(\frac{3\rho}{16\gamma_t} - \frac{25\kappa^2}{6\eta_t \mu \lambda}\right)\|x_{t+1} - x_t\|^2 + \frac{19\gamma_t}{8\rho}\|v_t - \nabla_x f(x_t, y_t)\|^2$$

$$- \left(\frac{\eta_t \mu \lambda}{4} - \frac{19 L_f^2 \gamma_t}{8\rho}\right)\|y_t - y^*(x_t)\|^2 - \frac{3\eta_t}{4}\|\tilde{y}_{t+1} - y_t\|^2 + \frac{25\eta_t \lambda}{6\mu}\|\nabla_y f(x_t, y_t) - w_t\|^2$$

$$\leq -\frac{3\gamma_t \rho}{32}\|\mathcal{G}_t\|^2 - \left(\frac{3\rho}{16\gamma_t} - \frac{25\kappa^2}{6\eta_t \mu \lambda}\right)\|x_{t+1} - x_t\|^2 + \frac{19\gamma_t}{8\rho}\|v_t - \nabla_x f(x_t, y_t)\|^2$$

$$- \frac{3\eta_t}{4}\|\tilde{y}_{t+1} - y_t\|^2 + \frac{25\eta_t \lambda}{6\mu}\|\nabla_y f(x_t, y_t) - w_t\|^2, \tag{94}$$

where the last inequality is due to $0 < \gamma \leq \frac{\eta_t \mu \lambda \rho}{38 L_f^2}$.

Let $\gamma = \gamma_t$ and $\eta = \eta_t$ for all $t \geq 1$. Thus, we have

$$\frac{3\gamma\rho}{32}\mathbb{E}\|\mathcal{G}_t\|^2 \leq \mathbb{E}\left[\Omega_t - \Omega_{t+1}\right] - \left(\frac{3\rho}{16\gamma} - \frac{25\kappa^2}{6\eta\mu\lambda}\right)\mathbb{E}\|x_{t+1} - x_t\|^2 + \frac{19\gamma}{8\rho}\mathbb{E}\|v_t - \nabla_x f(x_t, y_t)\|^2$$

$$- \frac{3\eta}{4}\mathbb{E}\|\tilde{y}_{t+1} - y_t\|^2 + \frac{25\eta\lambda}{6\mu}\mathbb{E}\|\nabla_y f(x_t, y_t) - w_t\|^2. \tag{95}$$

Summing over $t = 1, 2, \cdots, T$ on both sides of (95), by Lemma 7, we have

$$\frac{3\gamma\rho}{32}\sum_{t=1}^T \mathbb{E}\|\mathcal{G}_t\|^2 \leq \mathbb{E}\left[\Omega_1 - \Omega_{T+1}\right] - \left(\frac{3\rho}{16\gamma} - \frac{25\kappa^2}{6\eta\mu\lambda}\right)\sum_{t=1}^T \mathbb{E}\|x_{t+1} - x_t\|^2 - \frac{3\eta}{4}\sum_{t=1}^T \mathbb{E}\|\tilde{y}_{t+1} - y_t\|^2$$

$$+ \left(\frac{19\gamma}{8\rho} + \frac{25\eta\lambda}{6\mu}\right)\sum_{t=1}^T \left(\frac{L_f^2}{b_1}\sum_{i=(n_t-1)q}^{t-1}\left(\mathbb{E}\|x_{i+1} - x_i\|^2 + \mathbb{E}\|y_{i+1} - y_i\|^2\right) + \frac{\sigma^2}{b}\right)$$

$$\leq \mathbb{E}\left[\Omega_1 - \Omega_{T+1}\right] - \left(\frac{3\rho}{16\gamma} - \frac{25\kappa^2}{6\eta\mu\lambda}\right)\sum_{t=1}^T \mathbb{E}\|x_{t+1} - x_t\|^2 - \frac{3\eta}{4}\sum_{t=1}^T \mathbb{E}\|\tilde{y}_{t+1} - y_t\|^2$$

$$+ \left(\frac{19\gamma}{8\rho} + \frac{25\eta\lambda}{6\mu}\right)\sum_{t=1}^T \left(\frac{L_f^2 q}{b_1}\left(\mathbb{E}\|x_{t+1} - x_t\|^2 + \mathbb{E}\|y_{t+1} - y_t\|^2\right) + \frac{\sigma^2}{b}\right)$$

$$= \mathbb{E}\left[\Omega_1 - \Omega_{T+1}\right] - \left(\frac{3\rho}{16\gamma} - \frac{25\kappa^2}{6\eta\mu\lambda} - \frac{19\gamma L_f^2 q}{8\rho b_1} - \frac{25\eta\lambda L_f^2 q}{6\mu b_1}\right)\sum_{t=1}^T \mathbb{E}\|x_{t+1} - x_t\|^2$$

$$- \left(\frac{3\eta}{4} - \frac{19\gamma L_f^2 q\eta^2}{8\rho b_1} - \frac{25\lambda L_f^2 q\eta^3}{6\mu b_1}\right)\sum_{t=1}^T \mathbb{E}\|\tilde{y}_{t+1} - y_t\|^2 + \left(\frac{19\gamma}{8\rho} + \frac{25\eta\lambda}{6\mu}\right)\frac{T\sigma^2}{b}$$

$$\leq \mathbb{E}\left[\Omega_1 - \Omega_{T+1}\right] + \frac{3}{4\eta L_f^2}\frac{T\sigma^2}{b}, \tag{96}$$

where the second inequality holds by $\sum_{t=1}^T \sum_{i=(n_t-1)q}^{t-1}\left(\mathbb{E}\|x_{i+1} - x_i\|^2 + \mathbb{E}\|y_{i+1} - y_i\|^2\right) \leq q\sum_{t=1}^T \left(\mathbb{E}\|x_{t+1} - x_t\|^2 + \mathbb{E}\|y_{t+1} - y_t\|^2\right)$; the third equality is due to $y_{t+1} = y_t + \eta_t(\tilde{y}_{t+1} - y_t)$; the last inequality is due to $b_1 = q$, $0 < \lambda \leq \frac{9\eta^2 L_f^2}{100\mu}$ and $0 < \gamma \leq \min\left(\frac{19}{3L_f^2\eta}, \frac{\rho\eta}{8}, \frac{9\rho\eta\mu\lambda}{400\kappa^2}\right)$, i.e., it easily be obtained from the following inequalities (97) and (98).

Since $b_1 = q$, $0 < \gamma \leq \frac{3\rho}{19L_f^2\eta}$ and $0 < \lambda \leq \frac{9\mu}{100\eta^2 L_f^2}$, we have $\frac{3\eta}{8} \geq \frac{19\gamma L_f^2 q\eta^2}{8\rho b_1}$ and $\frac{3\eta}{8} \geq \frac{25\lambda L_f^2 q\eta^3}{6\mu b_1}$, i.e., we obtain

$$\frac{3\eta}{4} \geq \left(\frac{19\gamma L_f^2 q\eta^2}{8\rho b_1} + \frac{25\lambda L_f^2 q\eta^3}{6\mu b_1}\right). \tag{97}$$

Thus, we also obtain $\frac{3}{4\eta L_f^2} \geq \left(\frac{19\gamma}{8\rho} + \frac{25\eta\lambda}{6\mu}\right)$ and $\frac{3}{4\eta} \geq \frac{19\gamma L_f^2 q}{8\rho b_1} + \frac{25\eta\lambda L_f^2 q}{6\mu b_1}$. Since $0 < \gamma \leq \min\left(\frac{3\rho}{19L_f^2\eta}, \frac{\rho\eta}{8}, \frac{9\rho\eta\mu\lambda}{400\kappa^2}\right)$, we have $\frac{3\rho}{32\gamma} \geq \frac{25\kappa^2}{6\eta\mu\lambda}$ and $\frac{3\rho}{32\gamma} \geq \frac{3}{4\eta}$. Thus, we have

$$\frac{3\rho}{16\gamma} \geq \frac{25\kappa^2}{6\eta\mu\lambda} + \frac{3}{4\eta} \geq \frac{25\kappa^2}{6\eta\mu\lambda} + \frac{19\gamma L_f^2 q}{8\rho b_1} + \frac{25\eta\lambda L_f^2 q}{6\mu b_1}. \tag{98}$$

By using the above inequality (96), we have

$$\frac{1}{T}\sum_{t=1}^{T}\mathbb{E}\|\mathcal{G}_t\|^2 \leq \frac{32\mathbb{E}(\Omega_1 - \Omega_{T+1})}{3\gamma\rho T} + \frac{8}{\gamma\rho\eta L_f^2}\frac{\sigma^2}{b}$$

$$= \frac{32(\tilde{F}(x_1) + \|y_1 - y^*(x_1)\|^2)}{3T\gamma\rho} - \frac{32\mathbb{E}(\tilde{F}(x_{T+1}) + \|y_{T+1} - y^*(x_{T+1})\|^2)}{3T\gamma\rho} + \frac{8}{\gamma\rho\eta L_f^2}\frac{\sigma^2}{b}$$

$$\leq \frac{32(\tilde{F}(x_1) - F^*)}{3T\gamma\rho} + \frac{32\Delta_1^2}{3T\gamma\rho} + \frac{8}{\gamma\rho\eta L_f^2}\frac{\sigma^2}{b}, \tag{99}$$

where the last inequality holds by Assumption 5 and $\Delta_1 = \|y_1 - y^*(x_1)\|$. According to Jensen's inequality, we have

$$\frac{1}{T}\sum_{t=1}^{T}\mathbb{E}\|\mathcal{G}_t\| \leq \left(\frac{1}{T}\sum_{t=1}^{T}\mathbb{E}\|\mathcal{G}_t\|^2\right)^{\frac{1}{2}} \leq \frac{4\sqrt{2(\tilde{F}(x_1) - F^*)}}{\sqrt{3T\gamma\rho}} + \frac{4\sqrt{2}\Delta_1}{\sqrt{3T\gamma\rho}} + \frac{2\sqrt{2}\sigma}{\sqrt{\gamma\rho\eta b}L_f}, \tag{100}$$

where the last inequality is due to the inequality $(a + b + c)^{\frac{1}{2}} \leq a^{\frac{1}{2}} + b^{\frac{1}{2}} + c^{\frac{1}{2}}$ for all $a, b, c \geq 0$.

$\square$