# OpenReview forum: "Efficient Mirror Descent Ascent Methods for Nonsmooth Minimax Problems"
_NeurIPS.cc/2021/Conference — NeurIPS 2021 Poster_

### Official Review · Reviewer_uFvW · 2021-07-15

**Rating:** 6
**Confidence:** 5

**Summary:**

This paper propose mirror descent ascent methods for nonsmooth nonconvex-strongly-concave minimax problem. The algorithms requires $O(\kappa\epsilon^{-2})$ gradient complexity for deterministic case and $O(\kappa^3\epsilon^{-3})$ stochastic gradient complexity for stochastic case.

**Limitations And Societal Impact:**

This is a theoretical paper.

**Main Review:**

This paper studies mirror descent for nonsmooth nonconvex-strongly-concave minimax problem. The algorithms integrate the mirror ascent iteration with the momentum technique to update $y$. The proposed mirror descent ascent algorithm requires $O(\kappa\epsilon^{-2})$ gradient complexity to find $\epsilon$-first-order stationary point and the complexity of variance-reduced mirror descent ascent algorithm is $O(\kappa^3\epsilon^{-3})$ stochastic gradient complexity.

The theoretical and empirical results of this paper is interesting, but comparison with some related work should be discussed in more details.
1. In deterministic case, Lin’s [A] algorithm could find $\epsilon$-stationary point with $\tilde O(\sqrt{\kappa}\epsilon^{-2})$ complexity, which is better than the algorithm in this paper.
2. In stochastic case, Luo's [20] algorithm also requires  $O(\kappa^3\epsilon^{-3})$ complexity. Although Luo's [20] algorithm does not consider constraint on $x$, it looks not difficult to be extended by introduced additional projection step on $x$.

I hope the authors address the above concerns in revision/feedback, which could make the paper more convincing.

[A] Lin, Tianyi, Chi Jin, and Michael I. Jordan. "Near-optimal algorithms for minimax optimization." Conference on Learning Theory. PMLR, 2020.

**Time Spent Reviewing:**

1

---

> ### Author Response · Authors · 2021-08-06
> **Responses for Comments**
>
> Thanks for your comments and suggestions. We address your concerns as follows:
>
> 1) In the **deterministic** case, from Theorem 2 in our paper ( lines 186-188 at the page 6), when given $\rho=O(L_f^{5/4})$ and $\lambda = O(\frac{1}{L_f})$, we have $\frac{1}{\gamma \rho }=O(\sqrt{\kappa})$ and our MDA method has a convergence rate of $O(\sqrt{\frac{\sqrt{\kappa}}{T}})$. Let $\sqrt{\frac{\sqrt{\kappa}}{T}}=\epsilon$,  our MDA method can also obtain sample (or gradient) complexity of $T=O(\sqrt{\kappa}\epsilon^{-2})$ for finding an $\epsilon$-stationary point, the same complexity as in [A]. Please note that Table 1 in our paper only shows that our MDA method has the complexity of $O(\kappa\epsilon^{-2})$ when given $\rho=O(L_f)$.
>
> In fact, the complexity $O(\sqrt{\kappa}\epsilon^{-2})$ in [A] is obtained by relying on the bounded constraint set $\mathcal{Y}$, i.e., $D_y=\max_{y, \ y’ \in \mathcal{Y}} ||y-y’|| < + \infty $ (Please see Theorem 12 at the page 11 in [A]). Thus, this complexity $O(\sqrt{\kappa}\epsilon^{-2})$ in [A] may not be applied to the unconstrained case in variable $y$, i.e., $\mathcal{Y}=\mathbb{R}^{p}$. However, our methods only assume that the constraint sets $\mathcal{X}$ and $\mathcal{Y}$ are convex. Thus, our methods and theoretical results can also be applied to the unconstrained minimax optimization ( i.e., $\mathcal{X}=\mathbb{R}^d$ and / or $\mathcal{Y}=\mathbb{R}^p$ ).
>
> 2) In the **stochastic** case, although our VR-SMDA algorithm and the SREDA algorithm [20] build on the same variance reduced technique of SIPIDER and have the same complexity of $O(\kappa^3\epsilon^{-3})$ for finding an $\epsilon$-stationary point, the SREDA algorithm relies on the Multi-Step gradient ascent of updating variable $y$ at each loop. However, our VR-SMDA algorithm only uses a single step gradient ascent of updating variable $y$ at each loop. Clearly, our VR-SMDA algorithm is simpler than the SREDA algorithm. At the same time, the SREDA algorithm only can use a small learning rate $\eta_k$ in updating variable $x$, which depends on a small $\epsilon$ (Please see Theorem 1 at page 6 in [20]). While our VR-SMDA algorithm can use a relatively large learning rate $\gamma_t$ in updating the variable $x$ without relying on the small $\epsilon$.
>
> Please note that Table 1 in our paper only shows that our VR-SMDA method has a sample (or gradient) complexity of $O(\kappa^3\epsilon^{-3})$ when given $\rho=O(L_f)$. In fact, when given $\rho=O(L_f^{3/2})$ and $\lambda = O(\frac{1}{\kappa L_f})$, we have $\frac{1}{\gamma\rho} = O(\kappa)$ and then our VR-SMDA has a convergence rate of $O(\sqrt{\frac{\kappa}{T}}+\sqrt{\frac{1}{\kappa b}})$. Let $\sqrt{\frac{\kappa}{T}} = \epsilon/2$, we have $T=O(\kappa\epsilon^{-2})$. At the same time, let $b_1=q=O(\epsilon^{-1})$ and $b=T/\kappa =O(\epsilon^{-2})$. Thus our VR-SMDA method  has a lower sample complexity of $b_1T+bT/q=O(\kappa\epsilon^{-3})$ for finding an $\epsilon$-stationary point.
>
> In the final version of our paper, we will add these discussions and cite the paper [A].

---

### Official Review · Reviewer_Mhm6 · 2021-07-15

**Rating:** 7
**Confidence:** 4

**Summary:**

This paper proposes a class of mirror descent ascent (MDA) methods for solving nonsmooth and nonconvex-strongly-concave minimax optimization problems by using dynamic mirror functions. It also studies the convergence properties of the proposed algorithms. In particular, the MDA method achieves a lower sample complexity by using the strongly-convex mirror function. The paper also provides some numerical experiments on fair classifiers and robust neural network training tasks to demonstrate the efficiency of the proposed algorithms. This paper first applies the mirror descent algorithm to nonsmooth minimax optimization. It provides a rigorous convergence analysis of the proposed algorithms. Overall, this paper is interesting and has some novelties. Thus I recommend accepting this paper.

**Limitations And Societal Impact:**

Yes

**Main Review:**

Some comments:
1) In the paper, problem (1) is a constrained optimization. Can the convergence results in the paper apply to the unconstrained minimax optimization?

2) In the convergence analysis, could the authors provide some intuitions about the defined Lyapunov function $\Omega_t$? Also, intuitively, why adopting mirror descent can help achieve a lower sample complexity?

3) In the experiment, how to choose the mirror functions used in the proposed algorithms?

Some typos:
1) In the line 85, “f(x;\xi^i)”  should be “\nabla f(x;\xi^i)”;
2) In the line 122, “f(x,y)=E[f(x;\xi)]”  should be  “f(x,y)=E[f(x,y;\xi)]”.

**Time Spent Reviewing:**

1

---

> ### Author Response · Authors · 2021-08-06
> **Responses for Comments**
>
> Thanks for your comments and suggestions. We address your concerns one by one as follows:
>
> 1) Yes, our convergence results can be applied to the unconstrained minimax optimization, since we only assume the constraint sets $\mathcal{X}$ and $\mathcal{Y}$ are convex.
>
> 2) For the Lyapunov function $\Omega_t$, the first term is the objective function on the variable $x$, which is a natural choice, and the section term is the gap between $y_t$ and $y^*(x_t)$, whose intuition comes from our Lemma 6. Our methods mainly use the strongly-convex Bregman functions to reduce the gradient (or sample) complexity of our methods.
>
> 3) In our experiments, we use the mirror (Bregman) functions $ \frac{1}{2}x^TH_tx $ for variable $x$ and $\frac{1}{2}y^TG_ty $ for variable $y$, respectively, where the matrices $H_t$ and $G_t$ are generated from the formulations (8) and (9) in our paper.
>
> Thanks for pointing out these typos. In our final version, we will correct them and proofread the whole paper.

---

### Official Review · Reviewer_SMH6 · 2021-07-16

**Rating:** 8
**Confidence:** 5

**Summary:**

This paper proposes efficient mirror descent ascent (MDA) methods to solve the nonsmooth nonconvex-strongly-concave minimax problems. The SOTA convergence rates are established both for deterministic and stochastic settings. In practice, some experimetal results are provided to demonstrate the efficiencies of the proposed MDA methods. Overall, this is a very good paper.

**Limitations And Societal Impact:**

Yes

**Main Review:**

This paper proposes a class of effective mirror descent ascent (MDA) methods to solve non-smooth non-convex strongly-concave minimax problems. It proves that the proposed MDA has a lower sample complexity than the best known results. At the same time, some experimental results on fair classifier and robust neural network training tasks verify the efficiency of the proposed algorithms.

The novelty of this paper lies in the use of the Bregman distance to reduce the sample complexity of the minimax optimization methods, which is very interesting in theory. Moreover, it provides a rigorous theoretical analysis. I only have some minor comments as follows.

1) Line 24: “f(x)” should be “h(y)”;
2) Line 85: “f(x;\xi^i)” should be “\nabla f(x;\xi^i)”;
3) Line 122: “f(x,y)=\mathbb{E}[f(x;\xi)]”  should be “f(x,y)=\mathbb{E}[f(x,y;\xi)]”;
4) Line 149, “mini-batch samples will take large variances in our SMDA algorithm” should be “our SMDA algorithm suffer from large variances due to only using mini-batch stochastic gradients”;
5) Line 213, “between our works and …” should be “between our methods...”;
6) In the experiment, how to choose these tuning parameters (such as \lambda, \gamma, \alpha) in the algorithms?

**Time Spent Reviewing:**

3

---

> ### Author Response · Authors · 2021-08-06
> **Responses for Comments**
>
> Thanks for your comments and suggestions. In our final version, we will correct all typos and proofread the whole paper. We choose the tuning parameters in our algorithms mainly based on our theoretical analysis. We will add more details on the parameter selection in our final version.

---

### Decision · Program_Chairs · 2021-09-27

**Decision:**

Accept (Poster)

**Comment:**

This paper considers nonconvex-strongly concave minimax optimization with Bregman divergence setup and provides several new results based on deterministic and stochastic mirror descent based algorithms. All the reviewers agree that the results in this paper are novel and interesting. The reviewers also raise some important concerns such as potential extension to concave but not strongly concave settings, which the authors should properly address in the camera ready version. I suggest to the authors to give serious thought to all other suggestions of the reviewers in the camera ready version, which will help this become an even stronger paper.